# Secure and Confidential Certificates of Online Fairness

**Olive Franzese**
University of Toronto & Vector Institute
Toronto, ON `olive.franzese@vectorinstitute.ai`

**Ali Shahin Shamsabadi**
Brave Software
London, UK

**Carter Luck**
University of Massachusetts, Amherst
Amherst, MA

**Hamed Haddadi**
Imperial College London & Brave Software
London, UK

## Abstract

The "black-box service model" enables ML service providers to serve clients while keeping their intellectual property and client data confidential. Confidentiality is critical for delivering ML services legally and responsibly, but makes it difficult for outside parties to verify important model properties such as fairness. Existing methods that assess model fairness confidentially lack either (i) ***reliability*** because they certify fairness with respect to a static set of data, and therefore fail to guarantee fairness in the presence of distribution shift or service provider malfeasance; and/or (ii) ***scalability*** due to the computational overhead of confidentiality-preserving cryptographic primitives. We address these problems by introducing ***online fairness certificates***, which verify that a model is fair with respect to data received by the service provider *online* during deployment. We then present OATH, a deployably efficient and scalable zero-knowledge proof protocol for confidential online group fairness certification. OATH exploits statistical properties of group fairness via a "cut-and-choose" style protocol, enabling scalability improvements over baselines.

## 1 Introduction

Service providers commonly adopt a "black-box service model" to provide ML-driven services to clients in sensitive domains like finance and healthcare. The confidentiality offered by the black box is critical for protecting data privacy and Intellectual Property (IP), but hinders the ability of outside parties to verify properties of the model such as fairness [16, 24]. Since ML models are susceptible to bias based on race, sex, color, disability, or location [43, 25, 45, 41, 29, 21, 6, 20, 2], verification of fairness is also critical in sensitive applications. In this work we reconcile these two important objectives–fairness and confidentiality–by asking:

> *How can we ensure that **deployed models** are fair while preserving **confidentiality**?*

Existing solutions (detailed in Appendix A) for confidential fairness assessment include black-box auditing [35, 34, 37] and cryptographic verification [40, 49, 28, 38, 44], but nearly all certify fairness *only* with respect to *offline data* such as a static training or audit dataset. Such *offline fairness certificates* suffer from unintentional or intentional fairness assessments failures when the model is deployed (Figure 1, first row).

**Unintentional issues of offline fairness certificates**. In practice, offline fairness certificates fail to generalize to unseen data encountered during deployment (e.g., *online client data*) as distribution shift is common in real-world applications [13, 4] and challenging to account for during training [11]. Distributional shifts can actively reinforce biases [9]. As a result, fairness guarantees established offline may not hold once the model is deployed. For example, a previous study of real-world

39th Conference on Neural Information Processing Systems (NeurIPS 2025).

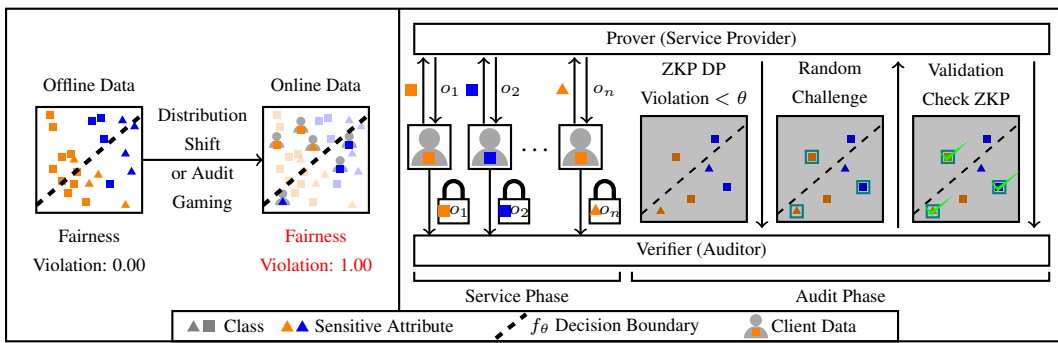

Figure 1: Limitations of Offline Fairness Certificates (left) and Overview of OATH (right). Offline fairness certificates based on offline audit data fail when facing real-world distribution shifts or audit gaming, leading to significant fairness deviation–Demographic Parity (DP) fairness violation increases from 0 to 1. OATH issues an online fairness certificate of the black-box service provided to clients reliably, efficiently, and securely. During the service phase, OATH authenticates client queries and service provider responses to enable accountability without having to perform client-facing ZKPs. In the audit phase, the service provider and an auditor verify only an asymptotically constant number of client queries while providing provable guarantees on overall group fairness violation.

distribution shift on US census data showed that fairness certified under distributions from 2014 failed to provide fairness for data gathered in 2018 [11].

**Intentional abuse of offline fairness certificates**. Many approaches for offline fairness certification [35, 34, 37, 9] can be circumvented entirely via a "model switching attack" wherein the service provider uses a fair model during the audit, and a different model during deployment. This attack allows arbitrary alterations to fairness on online client data, and can be concealed very effectively via post-hoc "fairwashing" methods [39, 1]. Service providers can bypass other audit methods [40] via a "data forging attack", wherein the malicious institution constructs an "easy" offline dataset on which the model satisfies fairness constraints, despite failing to meet those constraints for online data.

We formulate the concept of **online fairness certificates**, which remediate these issues by measuring fairness directly from the client data received online during deployment. To preserve confidentiality, we issue online fairness certificates using zero-knowledge proofs (ZKPs), a cryptographic method that enables verification of information about hidden data [18, 19].

Zero-knowledge proofs of fairness [40, 49] are an emerging line of work that address weaknesses of both black-box and white-box fairness audits using cryptographic tools. Black-box fairness auditing considers an auditor who has only query access to a model, and so measures fairness based on submitted queries and observed decisions of the service provider. By contrast, white-box fairness auditing approaches release client data and the service provider's model to the auditor, who can then verify fairness. Both approaches have drawbacks: the white-box setting is seldom used in practice due to issues with confidentiality, while the black-box setting does not provide the auditor enough information for a rigorous audit [9] (for example, nothing prevents a service provider from using one model during the audit, and a completely different one to answer client queries). Zero-knowledge proofs essentially provide secure "white-box" access to the auditor for a set of pre-specified operations, while retaining (and in fact improving upon) the confidentiality of the black-box setting.

We place two recent works in ZKP auditing into the newly defined category of online fairness certificates: [49] proves online certification of individual fairness, and [31] certifies general audit metrics. While these works are important, their deployment is substantially limited by a lack of scalability – ZKPs for validating ML services are computationally expensive, often requiring minutes of runtime to answer each client query [48, 49, 42]. Here we complement these works with a new method for online certification of group fairness [16, 22] with *excellent scalability*. Group fairness assesses statistical parity between demographic groups for outcome attribution and error rates. OATH achieves its scalability by exploiting a synergy between cryptographic "cut-and-choose"-style [53] verification and statistical properties of group fairness.

**Overview of OATH**. Our method consists of two phases (Figure 1, second row):

- *Service Phase.* Clients query the service provider's model, and send the auditor cryptographic commitments to their results. They perform no expensive ZKP operations, offloading them to the service provider and auditor in the next phase.

- *Audit Phase.* The service provider commits to a measurement of the fairness metric across all queries in the evaluation set. Then, they verify validity on a randomly sampled subset of the queries. This "cut-and-choose"-style [53] verification provides a statistical bound on the group fairness that is robust to arbitrary malicious behavior from the service provider, while maintaining excellent scaling for large numbers of client queries.

**Contributions.** We summarize our contributions as follows.

- *Online Group Fairness Certificate.* OATH audits group fairness over large sets of data received from clients online, rather than assuming fairness will generalize from a static set of offline data.

- *Scalability.* We exploit statistical properties of group fairness to audit an arbitrarily large set of queries to large neural networks using a *constant-sized* probabilistic sample (Theorem 4.1). On neural networks with 42.5 million parameters, OATH achieves 4.4 seconds of amortized runtime per query, of which only 0.23 seconds require the client to be online.

- *Confidentiality & Reliability.* Our cryptographic protocols guarantee that i) the auditor learns no information about the evaluation data or model parameters; and ii) the service provider cannot tamper with the audit's measurement of fairness, except within a very small probability and effect size that do not impact practical use.

Our code is publicly available at `https://github.com/cleverhans-lab/oath-zk-online-fairness.git`.

## 2 Background, Preliminaries & Related Work

**ML Preliminaries**. In this work we focus on probabilistic binary classifiers. That is, we consider models that can be represented as mappings $M : \mathcal{X} \times \{0,1\}^k \mapsto \{0,1\}$, where $\mathcal{X}$ is a feature space, and $\{0,1\}^k$ for some $k \in \mathbb{N}$ is the space of random seeds. We assume that one feature of each query point $q \in \mathcal{X}$ corresponds to a binary demographic attribute $\{0,1\} \in S$ such as sex, race, or disability.

**Fairness**. The ML community has proposed various fairness definitions tailored to different philosophical assumptions and contexts. We focus on *group fairness* [22, 12, 22] which ensures statistical parity across different subgroups. For simplicity, we demonstrate verification of demographic parity [7] in the main text, and generalize to other metrics in Appendix B. In this work we are interested in verifying whether demographic parity is satisfied within a public threshold, as formalized below.

**Definition 2.1.** *[Thresholded demographic parity] A predictor $\hat{Y}$ satisfies demographic parity with respect to sensitive attribute $S$ within a public threshold $\theta$ if:*

$$\left| \Pr[\hat{Y} = 1 | S = 0] - \Pr[\hat{Y} = 1 | S = 1] \right| \leq \theta$$

Demographic parity equalizes the probability of providing a positive outcome, $\hat{Y} = 1$ across each subgroups with different demographic variables 0 and 1. Therefore, receiving positive outcomes is independent of inclusion in a particular subgroup. This fairness definition is necessary for several life-changing tasks such as loan approval and recruitment so that applicants from different demographic groups have equal access to financial services and have equal chances of being hired [30].

**Zero-Knowledge Proofs** (ZKPs). A ZKP [18, 19] is a cryptographic protocol conducted between a *prover* $\mathcal{P}$ and a *verifier* $\mathcal{V}$, who both have access to a public circuit $P$. A ZKP allows $\mathcal{P}$ to convince $\mathcal{V}$ that they know some *witness* $w$ such that $P(w) = 1$. See Appendix C for formal properties of ZKPs. We use two ZKP protocols from prior work as building blocks for general zero-knowledge boolean circuit evaluation [47], and verified array random-access [17] respectively. The security of our methods follow directly from the security of these underlying building blocks, which are proven secure under the Universal Composability [8] framework.

We use the notation $[\![x]\!]$ to indicate that $\mathcal{P}$ and $\mathcal{V}$ hold an IT-MAC-based authentication of $x$. More precisely, it means that $\mathcal{V}$ holds a global key $\Delta$ and a message key $K_x$, while $\mathcal{P}$ holds a message tag $M_x$ and the value $x$. These components hold the algebraic relationship

$$M_x = K_x + x \cdot \Delta$$

as an invariant.

In this work, whenever an operation on authenticated values is written (e.g. $[\![z]\!] \leftarrow [\![x]\!] + [\![y]\!]$), it means that $\mathcal{P}$ and $\mathcal{V}$ are *working together* to perform special secure operations on their respective pieces of the authenticated values (detailed in [47, 17, 46]), such that the algebraic relationship is maintained. The relationship is used at the end of the ZKP protocol to check whether $\mathcal{P}$ performed computation accurately, without the $\mathcal{V}$ learning any information about the underlying values. See Appendix C for additional details.

**Commitments**. We employ a standard hash-based commitment scheme [27], which enables a party with an input value $x$ to produce a commitment string $C_x$ that can later be opened to verify the committed value. This prevents $\mathcal{P}$ from modifying client queries between the Service and Audit phases of our protocol. See Appendix C for details.

**ZKPs of Correct Inference**. A subset of ZKP methods are devoted to efficiently proving that inference was computed correctly given an ML model [48, 10]. We note that while their efficiency is improving substantially, they are still not fit for client-facing usage in many applications, requiring over ten minutes per verified inference for larger models. We use ZKPs of inference generically as a subroutine in our work as a component of proving fairness, conducted while the client is offline.

**Cryptographic Protocols and Fairness**. A few previous works leveraged ZKPs and secure multiparty computation (MPC) to enforce model fairness. Confidential-PROFITT [40] provides an efficient ZKP protocol for proving that a decision tree adheres to group fairness constraints at training time. However, Confidential-PROFITT suffers from issues of offline fairness certificates. FairProof [49], the work most closely related and complementary to ours, introduces online individual fairness [16, 26] certificates. However, FairProof requires several minutes of runtime per client query even for very small models. [28] extended MPC protocols proposed by [32] to enable a service provider to train a fair logistic regression model on clients' data while keeping their demographic attributes confidential. [38] uses MPC to verify fairness of inference. These MPC-based approaches require secure computation between the service provider and *every* client, which limits scalability. Our work achieves greatly improved efficiency by having the service provider *commit* to queries, and prove their validity probabilistically when the clients are offline.

In concurrent work, [52] proposed a new zkSNARK-based proof of fairness. While their approach is highly efficient, it relies on proving bespoke bounds for each new model type. In contrast, our work can "plug and play" to automatically prove fairness for any model with a zero-knowledge proof of inference. Thus while their results on logistic regression and multi-layered perceptrons are laudable, our work differs in its ability to support proofs of fairness for more complex architectures such as convolutional neural networks as we show in the results section.

See Appendix D for a recap of notation used in the paper.

## 3   Problem Formulation

**Parties and Inputs**. We consider three classes of parties: a service provider or *prover* $\mathcal{P}$, and a third-party auditor or *verifier* $\mathcal{V}$, and a set of *clients* $\{\mathcal{C}_i\}_{i=1}^n$ who possess the online data. Specifically, each client has a query point $q_i \in \mathcal{X}$, which they send to $\mathcal{P}$ in order to receive a binary decision $o_i \in \{0, 1\}$ from model $M$.

**Goal**. The auditor $\mathcal{V}$ aims to assess whether $\mathcal{P}$'s model is fair. Formally, the auditor wants to check whether $M$ satisfies Definition 2.1, given some public threshold $\theta$ on the fairness gap (set e.g. by regulations as an acceptable fairness gap in practice), measured via empirical probabilities over the online data and model decisions.

**Security Model**. Our protocols are secure against a malicious adversary corrupting $\mathcal{P}$ or $\mathcal{V}$. The adversary may perform arbitrary behavior during protocol execution to disrupt it. Nevertheless it

is guaranteed that (i) $\mathcal{P}$ cannot falsely convince $\mathcal{V}$ that the model is fair, except with negligible probability and effect size (characterized in Theorem 4.1), and (ii) $\mathcal{V}$ learns no information about any data or model parameters other than whether the fairness condition holds.

We assume that each client submits query points that are representative of real features and demographic attributes. Authentication of this information is typically placed outside of scope for algorithmic techniques. Instead it is delegated to outside mechanisms (e.g. the financial system authenticates inputs to loan recommendation models via legal liability for perjury). For clarity of notation in this work, we assume that each $\mathcal{C}_i$ submits a single query point $q_i \in \mathcal{X}$, though in practice clients could submit multiple queries.

We assume that none of $\mathcal{P}$, $\mathcal{V}$, or $\{\mathcal{C}_i\}_{i=1}^n$ collude with each other. Severe legal penalties prevent collusion between regulatory bodies and companies, making this reasonable in practice. We assume that all parties have access to secure point-to-point communication channels, and that all $\mathcal{C}_i$'s have been authenticated before protocol initiation – i.e. neither $\mathcal{P}$ nor $\mathcal{V}$ can launch "Sybil" attacks wherein they generate fake clients. We operate under the random oracle model for cryptographic hashing.

**Blame Attribution and Denial of Service**. A malicious $\mathcal{P}$ or $\mathcal{C}$ may tamper with the message sent to $\mathcal{V}$ during the Service Phase, which could be used to launch a denial of service attack. To deter this, we guarantee that if the Audit Phase aborts, either $\mathcal{P}$ or $\mathcal{C}$ can reveal information which correctly identifies the party responsible for the failure, preventing repeated denial of service.

# 4 OATH

The goal of our method is to enable a third-party auditor $\mathcal{V}$ to measure the fairness of the service provider $\mathcal{P}$'s ML model from online data provided by the clients $\{\mathcal{C}_i\}_{i=1}^n$. We use ZKPs to guarantee correct measurement of group fairness, while ensuring that $\mathcal{V}$ learns no information about the data or model parameters. OATH is made computationally efficient by its "cut-and-choose"-style [53] design: during the Service Phase, clients send the auditor binding and hiding cryptographic commitments to the online data and output they received from the service provider. Then in the Audit Phase, the auditor and service provider perform a ZKP to (i) measure group fairness from all online data, and then (ii) check a group-balanced random sample of the online data for tampering.

**Example Application.** Consider the case of algorithms deployed in criminal justice. Suppose $\mathcal{P}$ is the service provider of a proprietary ML model used in courtrooms to assign risk scores to defendants. A set of $N$ court offices utilizing the model are the clients $\{C_i\}_{i=1}^N$, and an independent review body seeking to measure fairness of the algorithm is the auditor $\mathcal{V}$.

To utilize OATH, in the Service Phase each court office collects dossiers from the defendants. The offices submit the dossiers as queries to $\mathcal{P}$, and send a cryptographic 'receipt' of each query to $\mathcal{V}$ via Algorithm 1. Then in the Audit Phase, $\mathcal{P}$ and $\mathcal{V}$ use Algorithm 2 to prove group fairness across all received queries. Using OATH in this application accomplishes three main goals: (i) the fairness of $\mathcal{P}$'s model is measured correctly – $\mathcal{P}$ cannot alter their outputs to make the model appear more fair, (ii) intellectual property is protected since neither the court offices nor $\mathcal{V}$ learn anything about the model, and (iii) data privacy is protected since the review body obtains no information about the defendants, which is important in many real-world settings where legislation may prevent the sharing of data across jurisdictions.

## 4.1 Service Phase

The first part of OATH is the Service Phase, wherein each $\mathcal{C}_i$ queries $\mathcal{P}$'s model, and $\mathcal{V}$ receives a cryptographic commitment to each answer. These commitments leak no information to $\mathcal{V}$. They will later be used in the Audit Phase to verify the correctness of the fairness audit via a ZKP.

Algorithm 1 performs committed query answering. First, $\mathcal{C}$ and $\mathcal{P}$ generate fair random coins (obtained using e.g. [5]) to prevent manipulation of the randomness used for the model decision. $\mathcal{C}$ samples two random strings $\alpha_0, \alpha_1$ to encode their sensitive attribute $s$. They send both $\alpha_0, \alpha_1$ to $\mathcal{V}$, and only $\alpha_s$ to $\mathcal{P}$. We note that the purpose of $\alpha_s$ is *not* to hide $s$ from $\mathcal{P}$ – $s$ is a component of $q$ which $\mathcal{P}$ gets to see in plain text. Rather, these values prevent $\mathcal{P}$ from being able to change $s$ before the audit without getting caught.

---

**Algorithm 1:** OATH Committed Query Answering.

---

**Input:** Public: security parameter $\lambda$; $\mathcal{C}$: query point $q$; $\mathcal{P}$: model $M$; $\mathcal{V}$: no inputs.
**Output:** $\mathcal{C}$: model decision $o$, randomness $r$; $\mathcal{P}$: query point $q$, randomness $r$; $\mathcal{V}$: commitment string
$\quad\quad C = H(q||o||r)$.

1   $\mathcal{C}$ and $\mathcal{P}$ generate fair coins $r$;
2   $s \leftarrow q.\text{sensitive\_attribute} \in \{0,1\}$ ;
3   $\mathcal{C}$ samples $\alpha_0, \alpha_1 \xleftarrow{\$} \{0,1\}^\lambda$;
4   $\mathcal{C}$ sends $q||\alpha_s$ to $\mathcal{P}$;
5   $\mathcal{C}$ computes $\text{sig}_P \leftarrow \text{Sign}(q||\alpha_s||r)$ and sends $\text{sig}_P$ to $\mathcal{P}$;
6   $\mathcal{P}$ computes $\text{Vrfy}(q||\alpha_s||r, \text{sig}_P)$. Abort if it fails;
7   $\mathcal{P}$ computes $o \leftarrow M(q, r)$ and $\text{sig}_C \leftarrow \text{Sign}(q||\alpha_s||o||r)$;
8   $\mathcal{P}$ sends $o, \text{sig}_C$ to $\mathcal{C}$;
9   $\mathcal{C}$ computes $\text{Vrfy}(q||\alpha_s||o||r, \text{sig}_C)$. Abort if it fails;
10   $\mathcal{C}$ computes $C \leftarrow H(q||\alpha_s||o||r)$, and sends $C$ and $\alpha_0, \alpha_1$ to $\mathcal{V}$.

---

In the Audit Phase, $\mathcal{P}$ will prove in zero-knowledge that the string $\alpha_s$ matches the sensitive attribute they received from $\mathcal{C}$. $\mathcal{P}$ can only guess $\alpha_{\neg s}$ with negligible probability, so this prevents them from flipping the client's sensitive attribute bit before the Audit Phase. This enables an efficient sensitive attribute check in Algorithm 2.

$\mathcal{C}$ then sends a signed query to $\mathcal{P}$, and $\mathcal{P}$ responds with a model output only if they successfully verify the signature. Likewise, $\mathcal{P}$ sends to $\mathcal{C}$ a signed query answer, which the client verifies before accepting the output. If both steps succeed, $\mathcal{C}$ sends a commitment to the query and result to $\mathcal{V}$. $\mathcal{C}$ also sends $\alpha_0, \alpha_1$, the random strings encoding

**Blame Attribution**. If verification fails in the Audit Phase, the signature on each party's inputs provides a publicly verifiable record of the query which can be revealed to $\mathcal{V}$ to identify which party was the source of the dishonest behavior. See Appendix E for technical details.

**Online Efficiency**. Cryptographic commitments are much less computationally intensive than ZKPs. By supplying $\mathcal{V}$ with commitments that can be verified via ZKP without any client input, we obtain client-facing efficiency comparable to ML as a service query answering without cryptographic verification. By contrast, performing a ZKP of fair inference online with each client requires runtimes on the order of minutes per client with state of the art methods, even on tiny neural networks [49]. Our commit online, prove offline design dramatically improves usability and scalability.

At the end of the Service Phase, $\mathcal{V}$ will have obtained a commitment $C_i$ and sensitive attribute check strings $\alpha_0^i, \alpha_1^i$ corresponding to the query and answer of each $\{\mathcal{C}_i\}_{i=1}^n$. In the Audit Phase, these committed queries are assessed for fairness.

## 4.2   Audit Phase

The second phase of OATH measures group fairness from committed online data via a suite of ZKP operations conducted solely between $\mathcal{P}$ and $\mathcal{V}$ while $\mathcal{C}$ stays offline. Our protocols guarantee correctness of the fairness measurement, and prevent $\mathcal{V}$ from learning any additional information about the data or model parameters.

Algorithm 2 begins with $\mathcal{P}$ making IT-MAC-based authentications of all query-answer tuples $(q_i, r_i, o_i) \in Q$. This enables assessment of fairness in lines 6-11, via verified estimation of $\Pr[o_i = 1|s_i = 0]$ and $\Pr[o_i = 1|s_i = 1]$ (line 11) on all queries in $Q$. Line 12 samples $\nu$ queries uniformly from each group (i.e. $\nu$ with $s_i = 0$ and $\nu$ with $s_i = 1$). Algorithm 6 details a ZKP subroutine for group-balanced sampling, placed in Appendix F for brevity.

**Validity Checks**. We implement three validity checks to deter $\mathcal{P}$ from modifying $Q$:

- *Sensitive Attribute Check* – $\mathcal{P}$ fails if they try to modify the sensitive attribute of any client.

- *Commitment Consistency Check* – $\mathcal{P}$ fails if they modified any part of the query sent by $\mathcal{C}_i$ during the Service Phase, for all $\mathcal{C}_i$ selected by the group-balanced uniform sample.

**Algorithm 2:** OATH Zero-Knowledge Fairness Audit.

---

**Input:** *public*: the number of client queries $n$, fairness gap threshold $\theta$, soundness parameter $\nu$; $\mathcal{P}$: model $M$, online data $Q = \{(q_i, \alpha_s^i, o_i, r_i)\}_{i=1}^n$; $\mathcal{V}$: commitments $\{C_i\}_{i=1}^n$, sensitive attribute check strings $\{(\alpha_0^i, \alpha_1^i)\}_{i=1}^n$

**Output:** $\mathcal{V}$ obtains $b_{\text{pass}} \in \{0, 1\}$ indicating whether $M$ satisfies demographic parity with respect to $Q$.

`// Step 1: Initialization`

1 **for** $i \in [1, n]$ **do**

2 $\quad$ $\mathcal{P}$ authenticates $(\llbracket q_i \rrbracket, \llbracket \alpha_s^i \rrbracket, \llbracket r_i \rrbracket, \llbracket o_i \rrbracket)$;

3 $\mathcal{P}$ authenticates $\llbracket M \rrbracket$;

4 $\mathcal{P}$ authenticates $\llbracket c_0 \rrbracket$ and $\llbracket c_1 \rrbracket$ initialized to zero; $\quad \triangleright$ Count positive outcomes in each demographic group

5 $\mathcal{P}$ authenticates $\llbracket n_0 \rrbracket$ and $\llbracket n_1 \rrbracket$ initialized to zero; $\quad \triangleright$ Count individuals in each demographic group

`// Step 2: Measuring Group Fairness`

6 **for** $i \in [1, n]$ **do**

7 $\quad$ $\llbracket s_i \rrbracket \leftarrow \llbracket q_i.\text{demographic\_attribute} \rrbracket$;

8 $\quad$ $\llbracket b_0 \rrbracket \leftarrow (\llbracket s_i \rrbracket == 0), \quad \llbracket b_1 \rrbracket \leftarrow (\llbracket s_i \rrbracket == 1);$ $\quad \triangleright$ Indicator bit for demographic attribute

9 $\quad$ $\llbracket n_0 \rrbracket \leftarrow \llbracket n_0 \rrbracket + \llbracket b_0 \rrbracket, \quad \llbracket n_1 \rrbracket \leftarrow \llbracket n_1 \rrbracket + \llbracket b_1 \rrbracket;$ $\quad \triangleright$ Update group counts

10 $\quad$ $\llbracket c_0 \rrbracket \leftarrow \llbracket c_0 \rrbracket + (\llbracket b_0 \rrbracket \cdot \llbracket o_i \rrbracket), \quad \llbracket c_1 \rrbracket \leftarrow \llbracket c_1 \rrbracket + (\llbracket b_1 \rrbracket \cdot \llbracket o_i \rrbracket);$ $\quad \triangleright$ Update positive outcome counts

11 Fairness gap computation and comparison to the threshold $\llbracket b_{\text{pass}} \rrbracket \leftarrow \left( \theta \geq \left| \frac{\llbracket c_0 \rrbracket}{\llbracket n_0 \rrbracket} - \frac{\llbracket c_1 \rrbracket}{\llbracket n_1 \rrbracket} \right| \right)$;

`// Step 3: Validity Checks`

12 $S \leftarrow$ Group-Balanced Uniform Sampling$(Q, \nu)$; $\quad \triangleright$ An array indicating selected samples (Algorithm 6)

13 **for** $i \in [1, n]$ **do**

$\quad$ `// Sensitive Attribute Check`

14 $\quad$ $\mathcal{V}$ sends $\alpha_0^i, \alpha_1^i$ to $\mathcal{P}$;

15 $\quad$ $\mathcal{P}$ proves $\left( (\llbracket \alpha_s^i \rrbracket == \alpha_0^i) \oplus (\llbracket \alpha_s^i \rrbracket == \alpha_1^i) \right) == 1$;

16 $\quad$ $\llbracket s_i' \rrbracket \leftarrow (\llbracket \alpha_s^i \rrbracket == \alpha_1^i)$;

17 $\quad$ $\mathcal{P}$ proves $\llbracket s_i \rrbracket == \llbracket s_i' \rrbracket$;

18 $\quad$ **if** `Reveal(`$\llbracket S[i] \rrbracket$`) == 1` **then**

$\quad\quad$ `// Commitment Consistency Check`

19 $\quad\quad$ $\mathcal{P}$ proves $H(q_i || \alpha_s^i || r_i || o_i) == C_i$;

$\quad\quad$ `// Inference Correctness Check`

20 $\quad\quad$ $\mathcal{P}$ proves $\llbracket o_i \rrbracket == \llbracket M \rrbracket(\llbracket q_i \rrbracket, \llbracket r_i \rrbracket)$ using $\mathcal{F}_{\text{inf}}$;

21 If any of the proofs fail, abort. Otherwise, `Reveal(`$\llbracket b_{\text{pass}} \rrbracket$`)`

---

- *Inference Correctness Check* – $\mathcal{P}$ fails if they gave $\mathcal{C}_i$ any output $o_i \neq M(q_i, r_i)$ during the Service Phase, for all $\mathcal{C}_i$ selected by the group-balanced uniform sample.

In line 15 $\mathcal{P}$ verifies that the $\llbracket \alpha_s^i \rrbracket$ authenticated in Step 1 is equal to at least one of the sensitive attributes committed to the client, without revealing which one. Since $\mathcal{C}_i$ only sends $\alpha_s^i$ to $\mathcal{P}$ in the Service Phase and $\mathcal{V}$ waits until *after* $\llbracket \alpha_s^i \rrbracket$ is authenticated to send $\alpha_0^i, \alpha_1^i$, a malicious $\mathcal{P}$ can only falsify this check if they guess $\alpha_{\neg s}^i$. This occurs only with negligible probability. This approach to verification is similar to the generation of random labels for wire values in garbled circuits, a classical cryptographic technique [50]. Lines 16 and 17 verify that the sensitive attribute encoded by $\llbracket \alpha_s^i \rrbracket$ is consistent with the sensitive attribute used in the rest of the protocol.

Line 19 verifies commitment consistency by proving that the values $\mathcal{P}$ used in the Audit Phase are consistent with the values that $\mathcal{C}_i$ committed to in the Service Phase. Line 20 verifies that $o_i$ was the proper result of inference with the authenticated model $M(q_i, r_i)$ by using a ZKP of correct inference $\mathcal{F}_{\text{inf}}$ as a subroutine. In line 21, the indicator bit $b_{\text{pass}}$ is revealed as long as the ZKP protocol was carried out correctly and all of the checks in lines 13-20 pass. This signals to $\mathcal{V}$ whether the fairness condition (Definition 2.1) holds, within a probabilistic bound characterized in the next section.

### 4.3 Analysis of Probabilistic Audit

To achieve highly efficient amortized runtime and communication, the zero-knowledge audit in Algorithm 2 performs a *probabilistic* check on audit integrity by verifying a random subset of queries sampled from $Q$. Our analysis shows that the probability with which a malicious $\mathcal{P}$ can wrongly demonstrate that their model is fair decreases exponentially as a function of sample size.

We proceed by defining an ideal measure of fairness – the group fairness gap that would be computed if $\mathcal{P}$ was fully honest – and comparing it to the group fairness gap computed when a malicious $\mathcal{P}$ submits a modified collection of queries as input to Algorithm 2. We then analyze the probability that $\mathcal{P}$ is able to execute Algorithm 2 without detection when these quantities differ by some $\epsilon > 0$.

**Definition 4.1.** *Honest & Measured Fairness Gaps. Let $(q_i, r_i, o_i) \in Q_h$ be the collection of query-answer tuples input to Algorithm 2 assuming a perfectly honest $\mathcal{P}$, which notably implies: i) queries are always answered using correct computations of inference $M$; and ii) no modifications have been made to the queries recorded in the Service Phase.*

*We will refer to $X_h$, demographic parity gap computed on $Q_h$, as the **honest fairness gap**. We have*

$$X_h = \left| \frac{\sum_{(q_i,r_i,o_i)\in Q_h} o_i \cdot I_a(s_i)}{\sum_{(q_i,r_i,o_i)\in Q_h} I_a(s_i)} - \frac{\sum_{(q_i,r_i,o_i)\in Q_h} o_i \cdot I_b(s_i)}{\sum_{(q_i,r_i,o_i)\in Q_h} I_b(s_i)} \right|.$$

*Where the indicator $I_a$ is 1 if $s_i == a$ and 0 otherwise.*

*Let $X_m$, the **measured fairness gap**, refer to the same quantity but computed on an arbitrary collection of query-answer tuples of a malicious $\mathcal{P}$'s choosing $Q_m$.*

$$X_m = \left| \frac{\sum_{(q_i,r_i,o_i)\in Q_m} o_i \cdot I_a(s_i)}{\sum_{(q_i,r_i,o_i)\in Q_m} I_a(s_i)} - \frac{\sum_{(q_i,r_i,o_i)\in Q_m} o_i \cdot I_b(s_i)}{\sum_{(q_i,r_i,o_i)\in Q_m} I_b(s_i)} \right|.$$

**Theorem 4.1.** *Let $X_h$ be the honest group fairness gap and $X_m$ the measured group fairness gap computed during Algorithm 2. Consider $\epsilon > 0$, the deviation between these quantities caused by a malicious $\mathcal{P}$ cheating, defined: $\epsilon = |X_h - X_m|$. Then $\mathcal{P}$ is caught with probability*

$$p_{catch} \geq 1 - \left(1 - \frac{\epsilon}{2}\right)^{\nu},$$

*where $\nu$ is the number of queries uniformly sampled within each group via Algorithm 6 (a total of $2\nu$).*

The proof is in Appendix G. The intuition is that the larger $Q$ is, the more queries $\mathcal{P}$ must maliciously modify to alter the group fairness gap (Definition 4.1). This in turn increases the probability that a modified query shows up in the verified random sample, resulting in $\mathcal{P}$ getting caught.

Theorem 4.1 shows that the probability that $\mathcal{P}$ can escape detection for altering $\mathcal{V}$'s measurement of the group fairness gap declines exponentially as a function of the number of sampled queries $2\nu$. The base of the exponent increases with the magnitude of the alteration $\epsilon$. This bounds the probability that $\mathcal{P}$ escapes detection at any given $\nu$. Thus, we can parameterize the audit to make it highly improbable that $\mathcal{P}$ could impact the measured fairness by amounts that matter in practice. We conduct empirical analysis of parameter settings in Section 5.

## 5 Empirical Evaluation

**Objectives.** We empirically evaluate the efficiency, scalability, and correctness of OATH for providing a repeatable fairness audit of ML-based services while protecting the confidentiality of evaluation data and model parameters.

**Datasets.** We consider five common datasets for fairness benchmarking (described in Appendix H): COMPAS [2], Crime [36], Default Credit [51], Adult [3] and German Credit [15].

**Implementation.** We implement OATH in C++ using EMP-toolkit [46] (under an MIT License). All experiments were conducted by locally simulating the parties on a MacBook Pro laptop CPU.

**Models.** OATH uses a zero-knowledge proof of correct inference $\mathcal{F}_{inf}$ as a subroutine in order to accommodate arbitrary binary classifiers. We evaluate OATH using three different settings for $\mathcal{F}_{inf}$: (i) logistic regression (LR) implemented in EMP-toolkit, (ii) a small feed-forward neural network (FFNN) with ReLU activations suitable for tabular data implemented in EMP-toolkit, (iii) larger neural networks suitable for image data using Mystique [48] with three different networks LeNet-5 (62K parameters), ResNet-50 (23.5 mil parameters), and ResNet-101 (42.5 mil parameters).

**Baselines.** We compare the efficiency of OATH against a baseline method for confidential online fairness certificates of group fairness. The runtime for this baseline is estimated by (i) computing

Table 1: Mean & SD runtimes across 3 trials of the Service and Audit Phases of OATH when applied to a Logistic Regression (LR) or Feed-Forward Neural Network (FFNN). For the Service Phase, we report the runtime (s) per query for each client $\mathcal{C}_i$. For the Audit Phase, which occurs between the service provider ($\mathcal{P}$) and the auditor ($\mathcal{V}$), we report the total runtime (min) across $10^6$ client queries.

| Dataset | Service (s/query) | Audit (min) | |
| --- | --- | --- | --- |
| | | LR | FFNN |
| COMPAS | $0.25 \pm 0.04$ | $82.8 \pm 0.01$ | $93.1 \pm 0.29$ |
| Crime | $0.23 \pm 0.02$ | $83.3 \pm 0.02$ | $101.8 \pm 0.6$ |
| Credit | $0.23 \pm 0.03$ | $83.3 \pm 0.03$ | $103.2 \pm 1.0$ |
| Adult | $0.22 \pm 0.02$ | $83.1 \pm 0.02$ | $97.9 \pm 0.8$ |
| German | $0.23 \pm 0.03$ | $84.7 \pm 0.06$ | $123.3 \pm 1.4$ |

a ZKP of correct inference using $\mathcal{F}_{\text{inf}}$ for each query, and (ii) proving that the fairness metric is satisfied over all queries. We compute this baseline for LR, FFNN, and larger neural networks using Mystique [48]. We also compare to FairProof [49], a ZKP method for online certificates of *individual* fairness. We emphasize that individual fairness and group fairness are distinct metrics, and as such OATH and FairProof are performing *distinct, complementary* tasks. We compare against FairProof to give context for the computational overhead of online fairness certification.

## 5.1 Efficiency of Certifying Standard Fairness Benchmarks

OATH consists of a client-facing Service Phase, and an Audit Phase conducted solely between the service provider and auditor after many client queries have been aggregated. Table 1 shows the runtime of each phase for all five datasets across multiple runs. Overall, the run times are practically efficient even for very large numbers of queries.

**Service Phase Runtime**. The first column of Table 1 reports Service Phase runtime, the time required to provide a committed response to one client query using $\mathcal{P}$'s model. The Service Phase has a negligible and consistent runtime across all datasets ranging from 0.22 to 0.25 seconds. This consistent efficiency is due to the small amounts of communication involved in the Service Phase, making the bottleneck the round time rather than the amount of communication. This makes the runtime appear independent of the number of attributes and the type of model used by the prover across all the datasets in these experiments. *The excellent efficiency of the Service Phase means that OATH can be integrated into deployed ML pipelines without disrupting service.*

**Audit Phase Runtime**. The second and third columns of Table 1 report Audit Phase runtime when OATH is verifying either a LR or FFNN model respectively. This is the time required to audit the fairness of authenticated client query answers with our ZKP protocols. We assume $10^6$ total client queries, 7600 of which are randomly selected for consistency and correctness checks. This size of the random sample was identified empirically as a good tradeoff between efficient runtime and strong tamper protection (see Section 5.2 for details). Figure 2 presents an ablation study of runtimes for the different components of the Audit Phase for logistic regression. The Correctness Check (Algorithm 2 line 20) dominates the runtime. See Appendix I.1 for a detailed breakdown of Audit Phase runtime. Though the Audit Phase is more computationally intensive, it is *highly practical for periodic auditing.*

## 5.2 Parametrization of Probabilistic Audit

Section 4.3 proves an efficiency / reliability trade-off: the more verified queries $2\nu$, the smaller the probability that $\mathcal{P}$ can alter the audit's fairness measurement by $\epsilon$ without being caught. As shown in the previous section, the runtime bottleneck of OATH scales with the number of verified queries. Therefore, it is important to identify parameters that limit the required runtime for the consistency check while ensuring a reliable measurement of fairness.

For example, setting $\nu = 1000$ means that any $\epsilon \geq 0.01$ evades detection with probability at most $6.65 \times 10^{-3}$. For $\nu = 3800$, the same $\epsilon$ evades with probability at most $5.34 \times 10^{-9}$. We select this value as a good tradeoff between efficiency and reliability. See Figure 3 for possible $\epsilon$ deviations from an example group fairness threshold $\theta$ at varying numbers of verified queries. Figure 5 in Appendix I.2 gives additional details. *These results show that OATH is robust: a service provider that cheats with a larger than negligible $\epsilon$ will be caught by the auditor with high probability.*

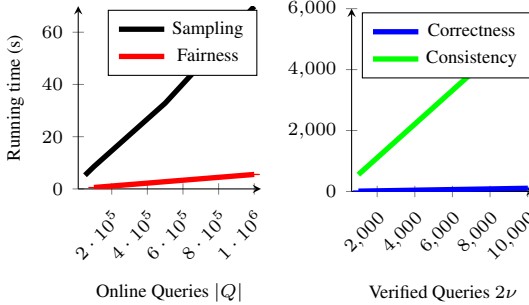

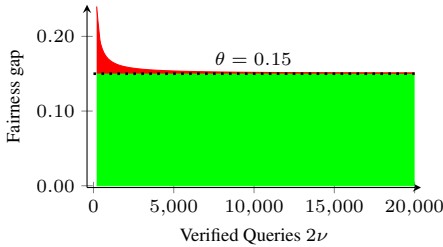

Figure 2: Ablation study of Audit Phase runtime for $\mathsf{OATH_{LR}}$: (left) group-balanced uniform sampling and fairness metric computation total online queries $|Q|$ varies; (right) correctness check and consistency check as a function of number of verified user queries.

Figure 3: Green: models underneath the example fairness threshold $\theta = 0.15$; Red: models placed over the by cheating which escape detection with greater than $1\%$ probability. The possible $\epsilon$ deviation that escapes detection decreases exponentially with number of verified queries by Theorem 4.1.

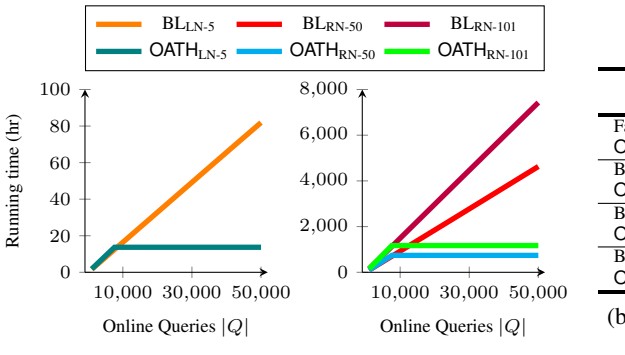

(a) Total Runtime

| | #Param | Time (s/query) | |
|---|---|---|---|
| | | Online | Total |
| FairProof | 130 | 236.4 | 316.8 |
| $\mathsf{OATH_{FFNN}}$ | 130 | **0.23** | **0.235** |
| Baseline | 62K | 5.9 | 5.9 |
| $\mathsf{OATH_{LN\text{-}5}}$ | 62K | **0.23** | **0.28** |
| Baseline | 23.5M | 333 | 333 |
| $\mathsf{OATH_{RN\text{-}50}}$ | 23.5M | **0.23** | **2.90** |
| Baseline | 42.5M | 535 | 535 |
| $\mathsf{OATH_{RN\text{-}101}}$ | 42.5M | **0.23** | **4.43** |

(b) Amortized runtime for $|Q| = 10^6$

Figure 4: Scalability of $\mathsf{OATH}$ compared to baseline with varying neural network sizes. Runtimes are estimated using ZKP inference times from [48]. The baseline approach uses ZKP verified inference with each client (as in [49, 31]) followed by verified group fairness computation over all queries.

## 5.3 Scalability for Neural Networks

Figure 4 shows experiments and estimated runtimes of $\mathsf{OATH}$ compared to neural network baselines. $\mathsf{OATH}$ is more efficient than the baselines by orders of magnitude. Especially important is the client-facing runtime, since this is arguably the most relevant factor for enabling the integration of ZKP auditing into ML pipelines. In this category, $\mathsf{OATH}$ gives at least a *thousand-fold* runtime improvement over the alternatives in three out of four categories, primarily due to the fact that it requires no client-facing ZKP. Even in terms of total amortized runtime, *$\mathsf{OATH}$ provides a substantial increase in efficiency in all categories due to our probabilistic auditing method.*

## 6 Conclusion

In this study we present $\mathsf{OATH}$, the first method for privacy-preserving certificates of online group fairness. In particular, we use zero-knowledge proofs which make it possible to measure fairness directly from evaluation data, thus removing onerous resource requirements for third-party auditors and the generalization error incurred by using validation sets. Further, our method makes dramatic improvements to scalability in comparison to baselines. See Appendix J for limitations. In future work, it may be interesting to extend cut-and-choose style ZKPs to obtain efficient proofs of other online properties, such as security, robustness, and/or calibration.

## Acknowledgements

Olive Franzese's work on this project was primarily supported by a Brave Software Internship. Olive Franzese was also supported by the National Science Foundation Graduate Research Fellowship Grant No. DGE-1842165. Resources used in preparing this research were provided, in part, by the Province of Ontario, the Government of Canada through CIFAR, and companies sponsoring the Vector Institute.

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

Table 2: Comparison of fairness auditing approaches in terms of fairness definition (individual; group; various), fairness certificate (online; offline), reliability and efficiency.

| Approach | Fairness | Certificate | Reliable | Efficient |
|---|---|---|---|---|
| Black-Box [34, 35, 37] | Various | Online | - | ✓ |
| Confidential-PROFIT [40] | Group | Offline | - | - |
| FairProof [49] | Individual | Online | ✓ | - |
| Fairness Baseline w/ Mystique [48] | Various | Online | ✓ | - |
| OATH | Group | Online | ✓ | ✓ |

## A    Comparison of fairness auditing approaches

We compared against existing fairness auditing approaches (summarised in Table 2), namely Black-Box audits [34, 35, 37] as well as both previous works that leveraged ZKPs to enforce model fairness (FairProof [49] and Confidential-PROFIT [40]). We also created a baseline based on ZKP of correct inference, Mystique [48].

Black-box fairness auditing considers an auditor who has only query access to a model, and so measures fairness based on submitted queries and observed decisions of the service provider. By contrast, white-box fairness auditing approaches release client data and the service provider's model to the auditor, who can then verify fairness. Both approaches have drawbacks: the white-box setting is seldom used in practice due to issues with confidentiality, while the black-box setting does not provide the auditor enough information for a rigorous audit [9] (for example, nothing prevents a service provider from using one model during the audit, and a completely different one to answer client queries). Zero-knowledge proofs of fairness [40, 49] are an emerging line of work that address both of these weaknesses using cryptographic tools. Zero-knowledge proofs essentially provide secure "white-box" access to the auditor for a set of pre-specified operations, while retaining (and in fact improving upon) the confidentiality of the black-box setting.

## B    Certifying Other Group Fairness Metrics

Here we present a modified version of the Audit Phase which measures equalized odds [22], another group fairness metric. Rather than measure the difference in positive outcomes between demographic groups as in demographic parity, equalized odds measures the difference in false positive and false negative rate between demographic groups.

**Definition B.1.** *[Thresholded equalized odds] A predictor $\hat{Y}$ satisfies equalized odds with respect to sensitive attribute $S$ within a public threshold $\theta$ if both of the following hold:*

$$|\Pr[\hat{Y} = 1 \mid y = 0 \cap s = a] - \Pr[\hat{Y} = 1 \mid y = 0 \cap s = b]| \leq \theta \qquad \forall a, b \in S$$

$$|\Pr[\hat{Y} = 1 \mid y = 1 \cap s = a] - \Pr[\hat{Y} = 1 \mid y = 1 \cap s = b]| \leq \theta \qquad \forall a, b \in S.$$

Computing equalized odds requires knowledge of the ground-truth labels for the data used. In certain applications such as finance, hiring, and medicine, ground truth labels may be readily available at audit time. For example, it is easy to determine whether a loan was repaid, whether an employee was promoted, or whether a patient's health metrics improved, at a later time point after the model makes a prediction. We abstract these sources of information via an oracle that gives access to ground truth labels, and present a version of OATH modified for equalized odds auditing in Algorithm 3. Other than the addition of the oracle in lines 4-5, it is highly similar to Algorithm 2, except that it computes empirical false positive and false negative rates for each group, which we abstract as a subroutine (Algorithm 4).

## C    Details of Cryptographic Preliminaries

### C.1    Properties of Zero-Knowledge Proofs

A zero-knowledge proof (ZKP) [18, 19] is a cryptographic protocol conducted between a *prover* $\mathcal{P}$ and a *verifier* $\mathcal{V}$, who both have access to a public circuit $P$. A ZKP allows $\mathcal{P}$ to convince $\mathcal{V}$ that they know some *witness* $w$ such that $P(w) = 1$. ZKPs have the following properties:

**Algorithm 3:** OATH Zero-Knowledge Equalized Odds Audit.

---

**Input:** *public*: the number of client queries $n$, fairness gap threshold $\theta$, soundness parameter $\nu$, ground truth oracle $\mathcal{O}$; $\mathcal{P}$: model $M$, evaluation data $Q = \{(q_i, \alpha_s^i, o_i, r_i)\}_{i=1}^n$; $\mathcal{V}$: commitments $\{C_i\}_{i=1}^n$, sensitive attribute check strings $\{(\alpha_0^i, \alpha_1^i)\}_{i=1}^n$

**Output:** $\mathcal{V}$ obtains $b_{\text{pass}} \in \{0, 1\}$ indicating whether $M$ satisfies demographic parity with respect to $Q$.

```
// Step 1:  Initialization
```
**1 for** $i \in [1, n]$ **do**
**2**    $\mathcal{P}$ authenticates $(\llbracket q_i \rrbracket, \llbracket \alpha_s^i \rrbracket, \llbracket r_i \rrbracket, \llbracket o_i \rrbracket)$;

**3** $\mathcal{P}$ authenticates $\llbracket M \rrbracket$;
**4 for** $i \in [1, n]$ **do**
**5**    $\llbracket g_i \rrbracket \leftarrow \mathcal{O}(\llbracket q_i \rrbracket)$

```
// Step 2:  Fairness Verification
```
**6** $\llbracket b_{\text{pass}} \rrbracket \leftarrow$ Equalized-Odds$(n, \theta, \llbracket Q \rrbracket, \{\llbracket g_i \rrbracket\}_{i=1}^n)$ ;
```
// Step 3:  Validity Checks
```
**7** $S \leftarrow$ Group-Balanced Uniform Sampling$(Q, \nu)$;        ▷ An array indicating selected samples
   (Algorithm 6)

**8 for** $i \in [1, n]$ **do**
```
   // Sensitive Attribute Check
```
**9**    $\mathcal{V}$ sends $\alpha_0^i, \alpha_1^i$ to $\mathcal{P}$ ;
**10**   $\mathcal{P}$ proves $\left((\llbracket \alpha_s^i \rrbracket == \alpha_0^i) \oplus (\llbracket \alpha_s^i \rrbracket == \alpha_1^i)\right) == 1$ ;
**11**   $\llbracket s_i' \rrbracket \leftarrow (\llbracket \alpha_s^i \rrbracket == \alpha_1^i)$ ;
**12**   $\mathcal{P}$ proves $\llbracket s_i \rrbracket == \llbracket s_i' \rrbracket$ ;
**13**   **if** *Reveal*$(\llbracket S[i] \rrbracket) == 1$ **then**
```
      // Commitment Consistency Check
```
**14**     $\mathcal{P}$ proves $H(q_i || \alpha_s^i || r_i || o_i) == C_i$;
```
      // Inference Correctness Check
```
**15**     $\mathcal{P}$ proves $\llbracket o_i \rrbracket == \llbracket M \rrbracket(\llbracket q_i \rrbracket, \llbracket r_i \rrbracket)$ using $\mathcal{F}_{\text{inf}}$;

**16** If any of the proofs fail, abort. Otherwise, Reveal$(\llbracket b_{\text{pass}} \rrbracket)$

---

- *Completeness*: For any $w$ such that $P(w) = 1$, $\mathcal{P}$ can use the ZKP protocol to convince an honest $\mathcal{V}$ that $P(w) = 1$;

- *Soundness*: Given $w$ such that $P(w) \neq 1$, a malicious $\mathcal{P}$ cannot use the ZKP protocol to falsely convince $\mathcal{V}$ that $P(w) = 1$;

- *Zero Knowledge*: The protocol reveals no information to even a malicious $\mathcal{V}$ about $w$ (other than what can be inferred from knowing that $P(w) = 1$).

### C.2 IT-MAC Authentication

We follow the methods for IT-MAC-based authentication in [47, 33, 14]. In more detail, we fix a field $\mathbb{F}_p$ for some prime $p \in \mathbb{N}$, and an extension field $\mathbb{F}_{p^r} \supseteq \mathbb{F}_p$. The notation $\llbracket x \rrbracket$ means that $\mathcal{V}$ holds a uniform global key $\Delta \in \mathbb{F}_{p^r}$ and for each authenticated value a uniform local key $\mathbf{K}_x \in \mathbb{F}_{p^r}$, while $\mathcal{P}$ holds the value $x \in \mathbb{F}_p$ and a uniform tag $\mathbf{M}_x \in \mathbb{F}_{p^r}$, over which the following algebraic relationship holds:

$$\mathbf{M}_x = \mathbf{K}_x + \Delta \cdot x \in \mathbb{F}_{p^r}$$

when for the purposes of the equality, $x$ is represented in the natural way as an element of $\mathbb{F}_{p^r}$. $\mathcal{P}$ can "open" an IT-MAC authenticated value by sending $x$ and $\mathbf{M}_x$ to $\mathcal{V}$, who can then verify that the relationship holds. If $\mathcal{P}$ has modified $x$ after authentication, it will only pass verification with negligible probability. The generic ZKP protocol in [47] works by authenticating the input values to a circuit, and then updating them in ways that preserve the algebraic relationship, which can only succeed if both $\mathcal{P}$ and $\mathcal{V}$ agree to perform the update (again, except with negligible probability). IT-MACs are *homomorphic* for addition and scalar multiplication, which means that if $\mathcal{P}$ and $\mathcal{V}$ agree, they can update authenticated values with these operations locally without any communication. This makes these operations very efficient in IT-MAC-based ZKPs.

**Algorithm 4:** OATH Equalized Odds Calculation.

---

**Input:** *public*: the number of client queries $n$, fairness gap threshold $\theta$; $\mathcal{P}$: authenticated evaluation data $[\![Q]\!] = \{([\![q_i]\!], [\![\alpha_s^i]\!], [\![o_i]\!], [\![r_i]\!])\}_{i=1}^n$, authenticated ground truth labels $G = \{[\![g_i]\!]\}_{i=1}^n$

**Output:** authenticated $[\![b_{\text{pass}}]\!] \in \{0, 1\}$ indicating whether the decisions in $Q$ satisfy demographic parity.

1  $\mathcal{P}$ authenticates $[\![P_0]\!], [\![P_1]\!], [\![N_0]\!]$ and $[\![N_1]\!]$ initialized to zero;     ▷ Count positive and negative outcomes in each demographic group

2  $\mathcal{P}$ authenticates $[\![TP_0]\!], [\![FP_0]\!], [\![TP_1]\!]$ and $[\![FP_1]\!]$ initialized to zero;     ▷ Count true and false positives in each demographic group

3  **for** $i \in [1, n]$ **do**

4      $[\![s_i]\!] \leftarrow [\![q_i.\text{demographic\_attribute}]\!]$;

5      $[\![b_0]\!] \leftarrow 1 - s_i, \quad [\![b_1]\!] \leftarrow s_i$;     ▷ Indicator bit for demographic attribute

6      $[\![P_0]\!] \leftarrow [\![P_0]\!] + ([\![b_0]\!] \cdot [\![g_i]\!]), \quad [\![P_1]\!] \leftarrow [\![P_1]\!] + ([\![b_1]\!] \cdot [\![g_i]\!])$ ;     ▷ Update positive counts

7      $[\![N_0]\!] \leftarrow [\![N_0]\!] + ([\![b_0]\!] \cdot (1 - [\![g_i]\!])), \quad [\![N_1]\!] \leftarrow [\![N_1]\!] + ([\![b_1]\!] \cdot (1 - [\![g_i]\!]))$ ; ▷ Update negative counts

8      $[\![TP_0]\!] \leftarrow [\![TP_0]\!] + ([\![b_0]\!] \cdot [\![o_i]\!] \cdot [\![g_i]\!]), \quad [\![TP_1]\!] \leftarrow [\![TP_1]\!] + ([\![b_1]\!] \cdot [\![o_i]\!] \cdot [\![g_i]\!])$;     ▷ Update true positive counts

9      $[\![FP_0]\!] \leftarrow [\![FP_0]\!] + ([\![b_0]\!] \cdot [\![o_i]\!] \cdot (1 - [\![g_i]\!])), \quad [\![FP_1]\!] \leftarrow [\![FP_1]\!] + ([\![b_1]\!] \cdot [\![o_i]\!] \cdot (1 - [\![g_i]\!]))$;     ▷ Update false positive counts

10  Fairness gap computation and comparison to the threshold

$$[\![b_{\text{pass}}]\!] \leftarrow \left( \theta \geq \left| \frac{[\![TP_a]\!]}{[\![P_a]\!]} - \frac{[\![TP_b]\!]}{[\![P_b]\!]} \right| \right) \cdot \left( \theta \geq \left| \frac{[\![FP_a]\!]}{[\![N_a]\!]} - \frac{[\![FP_b]\!]}{[\![N_b]\!]} \right| \right);$$

11  Return $[\![b_{\text{pass}}]\!]$

---

To provide a concrete example, we will briefly demonstrate how a secure addition is computed. Suppose $\mathcal{P}$ and $\mathcal{V}$ have authenticated values $[\![x]\!]$ and $[\![y]\!]$. This means that $\mathcal{P}$ holds $x, y$ and message tags $M_x$ and $M_y$, while $\mathcal{V}$ holds a global key $\Delta$ and message keys $K_x$ and $K_y$, with the algebraic relationships $M_x = K_x + \Delta \cdot x$ and $M_y = K_y + \Delta \cdot y$. To compute a secure addition, $\mathcal{P}$ and $\mathcal{V}$ must obtain $[\![z]\!]$ with the underlying value $z = x + y$ and a message key and message tag such that $M_z = K_z + \Delta \cdot z$.

Observe that by adding the respective components of $[\![x]\!]$ and $[\![y]\!]$ we can see that the following two equalities are equivalent

$$M_x + M_y = K_x + \Delta \cdot x + K_y + \Delta \cdot y$$
$$(M_x + M_y) = (K_x + K_y) + \Delta \cdot (x + y).$$

Thus $\mathcal{P}$ and $\mathcal{V}$ can achieve the goal of obtaining $[\![z]\!]$ as follows: $\mathcal{P}$ computes $z \leftarrow x + y$, and $M_z \leftarrow M_x + M_y$, while $\mathcal{V}$ computes $K_z \leftarrow K_x + K_y$.

We will also overload the double bracket notation to include collections of values: if $v$ is a vector and $A$ a matrix, $[\![v]\!]$ and $[\![A]\!]$ mean that each value in $v$ and $A$ is individually authenticated.

In summary, IT-MAC-based authentication provides a way to prevent $\mathcal{P}$ from modifying values except in ways that are approved by $\mathcal{V}$, even while keeping those values hidden. They also enable efficient computation with our ZKP building blocks [47, 17], however they require a global key $\Delta$ to be shared among all authenticated values for efficiency. This makes them cumbersome in applications with many parties. Accordingly, to prevent $\mathcal{P}$ from modifying the queries obtained by the many client parties, we use cryptographic commitments rather than IT-MAC authentication.

### C.3 Hash-Based Commitments

A commitment scheme enables a party with an input value $x$ to produce a commitment string $C_x$ which can later be opened to verify that $x$ has not changed. They possess the following properties:

- *Binding.* If $\mathcal{P}$ commits to $x$ and then modifies $x$ in any way without $\mathcal{V}$'s knowledge, verification of the commitment string will fail.
- *Hiding.* The commitment string reveals no information about $x$ to $\mathcal{V}$.

A standard approach [27] for instantiating a commitment scheme is to use a random oracle hash $H$, with

$$C_x := H(x||r)$$

giving the commitment string for any $x \in \{0,1\}^*$, where $r \in \{0,1\}^k$ is a randomly chosen bit string for some $k \in \mathbb{N}$. Verification is conducted by computing the hash again given $x$ and $r$, and confirming that it is equal to $C_x$ produced by the party giving the commitment. In this work, we use a ZKP protocol to confirm that $C_x = H(x||r)$ without having to send $x$ to $\mathcal{V}$. Thus we are able to verify that client queries have not been altered without revealing the queries to $\mathcal{V}$. We use a scheme from [48] to enable conversion of hash-based commitments to IT-MAC authentications within a ZKP protocol to enable this.

## C.4 Digital Signatures

A digital signature scheme guarantees that a signed message was sent by a sender with a known public key. Formally, a digital signature scheme consists of two algorithms:

- A key generation algorithm $\text{Gen}(1^\lambda) \to (sk, pk)$ which, given a security parameter generates a secret key and public key pair
- A signing algorithm $\text{Sign}_{sk}(m) \to \text{sig}$ which takes a secret key and message, and generates a signature
- A verification algorithm $\text{Vrfy}_{pk}(m, \text{sig}) \to \{0,1\}$ which takes a public key, a signature, and a message, and outputs a boolean value indicating whether the signature is valid (1 for valid, 0 for invalid).

A signature is valid if the pair $(sk, pk)$ was obtained from Gen, and sig was obtained from $\text{Sign}_{sk}(m)$. Informally, a secure signature scheme ensures that for any $m$, no adversary can forge sig' such that $\text{Vrfy}_{pk}(m, \text{sig}')$ outputs 1 unless they have access to the $sk$ corresponding to $pk$. See [27] for formal details.

## D Notation

Table 3 shows the notation used throughout this paper.

Table 3: Notation table.

| | Meaning | | Meaning |
|---|---|---|---|
| $\mathcal{P}$ | Prover | $\mathcal{V}$ | Auditor |
| $\mathcal{C}$ | Client | $M$ | Model |
| $D$ | Training Dataset | $Q$ | Evaluation dataset |
| $\theta$ | Group fairness gap threshold | $[\![x]\!]$ | Authentication of value $x$ |
| $S$ | Binary demographic attribute | $n$ | The number of client queries |
| $\mathcal{X}$ | Feature space | $\hat{Y}$ | Predictor |
| $q$ | Client's query | $P$ | Public circuit |
| $w$ | Witness | $o$ | Model binary decision |
| $\nu$ | Number of sampled queries from each demographic group | $\|$ | concatenation operator |
| $\oplus$ | XOR operator | | |

## E Details of Blame Attribution

If any of the proofs in Algorithm 2 fail, then the protocol aborts (line 21). To prevent denial of service attacks in real world applications, it is necessary to find out which party caused the failure. Notably, both the service provider $\mathcal{P}$ and the client $\mathcal{C}$ have incentives to deny service: a malicious $\mathcal{P}$ may seek to obscure the fact that they have a noncompliant model, and a malicious $\mathcal{C}$ may seek retribution for an undesirable model prediction. Here we formalize a method for correctly identifying which party is to blame if Algorithm 2 ends in an abort.

**Algorithm 5:** Blame Attribution

---

**Input:** Public: list of indices $I$ for all clients which failed validity checks in Algorithm 2, $\mathcal{P}$'s public key $pk$, list of sensitive attribute labels $L = \{(\alpha_0^i, \alpha_1^i)\}_{i \in |Q|}$, list of commitments $C = \{C_i\}_{i \in |Q|}$; $\mathcal{C}_i$: query $(q_i, \alpha_s^i, o_i, r_i)$ and $\mathcal{P}$ signature $\text{sig}_C^i$; $\mathcal{V}$ no inputs.

**Output:** a bit $b_i$ for each $i \in I$, 0 indicates $\mathcal{C}_i$ malfeasance, 1 indicates $\mathcal{P}$ malfeasance.

1 **for** $i \in I$ **do**
2 $\quad$ $\mathcal{C}_i$ reveals $(q_i, \alpha_s^i, o_i, r_i)$ and $\mathcal{P}$ signature $\text{sig}_C^i$ ;
3 $\quad$ $\mathcal{V}$ checks $\text{Vrfy}(q_i || \alpha_s^i || o_i || r_i, \text{sig}_C^i)$ using $\mathcal{P}$'s public key $pk$ ;
4 $\quad$ $\mathcal{V}$ checks $\text{Open}(q_i || \alpha_s^i || o_i || r_i, C_i)$. ;
5 $\quad$ $\mathcal{V}$ checks that $\alpha_s^i = \alpha_1^i$ or $\alpha_s^i = \alpha_0^i$ and that $s$ is equal to the sensitive attribute reported in $q_i$ ;
6 $\quad$ If any of the checks fail, $\mathcal{V}$ sets $b_i \leftarrow 0$, otherwise $b_i \leftarrow 1$

---

Since clients do not participate in the Audit Phase, they cannot interfere with any of the proofs before the validity checks (lines 13-20). Thus if protocol failure happens here, $\mathcal{P}$ is to blame. The interesting cases occur during the validity checks, where for some client $C_i$, $\mathcal{P}$ is verifying the sensitive attribute label $\alpha_s^i$, consistency of the commitment $C_i$, and correctness of the inference $o_i = M(q_i, r_i)$. If these proofs fail, then there are two possibilities: (i) $\mathcal{P}$ modified the $i^{th}$ entry of $Q$ between the Service Phase and the Audit Phase, or (ii) $\mathcal{C}_i$ submitted an invalid $\alpha_0^i, \alpha_1^i$ or $C_i$ to $\mathcal{V}$ during the Service Phase to intentionally disrupt the Audit Phase.

Algorithm 5 gives a protocol for post-hoc blame attribution. It relies on the security of the digital signature scheme used during the Service Phase (Algorithm 1): no malicious $\mathcal{C}$ can forge a signature $\text{sig}_C^i$ such that $\text{Vrfy}(q_i || \alpha_s^i || o_i || r_i, \text{sig}_C^i)$ will verify under $\mathcal{P}$'s public key (Appendix C.4). Thus, if line 3 succeeds it must indicate that $(q_i, \alpha_s^i, o_i, r_i)$ was exactly the query-response tuple given to $\mathcal{C}_i$ in line 8 of Algorithm 1. Thus if it matches up with the commitment $C_i$ given to $\mathcal{V}$ in line 10 of Algorithm 1, and $\alpha_s$ indexes the proper sensitive attribute, then $\mathcal{C}_i$ must have carried out Algorithm 1 honestly. This means that protocol failure in Algorithm 2 must have been caused by a malicious $\mathcal{P}$.

We note that Algorithm 5 requires $\mathcal{C}_i$ to reveal their data to $\mathcal{V}$. This can be remediated by instead using a zero-knowledge proof of digital signature verification, which has been explored in previous work. We reserve this possibility for future work.

## F ZKP Verification of Group-Balanced Uniform Sample

Algorithm 6 gives an interactive ZKP protocol for verifying that a sample contains $\nu$ uniform samples of points from each demographic group. Theorem G.2 characterizes its security guarantees.

## G Proof of Soundness

**Theorem G.1.** *Let $X_h$ be the honest group fairness gap and $X_m$ the measured group fairness gap computed during Algorithm 2. Consider $\epsilon > 0$, the deviation between these quantities caused by a malicious $\mathcal{P}$ cheating, defined:*

$$\epsilon = |X_h - X_m| \,.$$

*Then $\mathcal{P}$ is caught with probability*

$$p_{catch} \geq 1 - \left(1 - \frac{\epsilon}{2}\right)^{\nu},$$

*where $\nu$ is the number of queries uniformly sampled within each group via Algorithm 6 (a total of $2\nu$).*

*Proof.* We begin by considering available options for $\mathcal{P}$ to dishonestly influence $X_m$ (i.e. the quantity compared to $\theta$ in line 11 of Algorithm 2). Since any deviations influencing the computation of $c_0, c_1, n_0, n_1, S$ in lines 6-11 will be caught by the underlying ZKP protocols [47, 17], $\mathcal{P}$ can only cause a deviation in $X_m$ by behaving dishonestly to alter the query-answer tuples $(q_i, r_i, o_i)$ before they are committed during line 2.

This can happen in two ways: (i) by breaching *correctness* of the outcome given to $\mathcal{C}_i$ during Algorithm 1 (i.e. $o_i \neq M_\tau(q_i, r_i)$), or (ii) by breaching *consistency* (i.e. $(q_i, r_i, o_i)$ provided as input during Algorithm 2 line 2 $\neq (q, r, o)$ returned to $\mathcal{C}_i$ during Algorithm 1. A query-answer tuple breaching either condition will be caught by lines 18-20 of Algorithm 2 if it is sampled in $S$ during line 12.

Thus, in order to assess $\mathcal{P}$'s likelihood of being caught we must understand the number of modified queries required to produce a deviation between $X_h$ and $X_m$ of size $\epsilon$. Since the Sensitive Attribute Check (lines 14-17) causes the audit to fail if $\mathcal{P}$ modifies the sensitive attribute, we can proceed by just considering the case where $\mathcal{P}$ only modifies $o_i$ from $(q_i, r_i, o_i) \in Q_m$. Let $n_0$ be the number of query points in $Q_h$ such that $s_i == 0$, and let $n_1$ be defined similarly. Note that in the case where $\mathcal{P}$ only modifies $o_i$, these quantities are exactly the same when defined over $Q_m$.

Flipping any $o_i$ from $Q_h$ can produce at most $\frac{1}{n_0}$ alteration of $X_m$ if $s_i = 0$, or $\frac{1}{n_1}$ alteration if $s_i = 1$. This is because the numerator term changes by at most 1 per query. This means that in order to create a deviation of size $\epsilon$, $\mathcal{P}$ must modify at least $p_0 \cdot n_0 + p_1 \cdot n_1$ queries for some $p_0, p_1 \in [0, 1]$, where

$$\epsilon \leq p_0 + p_1.$$

This necessarily implies that either $p_0 \geq \frac{\epsilon}{2}$ or $p_1 \geq \frac{\epsilon}{2}$. Without loss of generality, assume the former.

Then if we uniformly sample $\nu$ queries with $s_i = 0$, the probability that *none* of them violate correctness or consistency (and $\mathcal{P}$ evades being caught) is given by

$$\Pr[\text{no modified queries in sample}] = (1 - p_0)^\nu$$
$$\leq (1 - \frac{\epsilon}{2})^\nu.$$

Since a symmetrical analysis holds for $p_1$, taking a $\nu$-sized uniform sample of queries from *both* groups (for a total of $2\nu$ verified queries) gives us the bound stated in the theorem.

Cases where $\mathcal{P}$ instead modifies $q_i$ or $r_i$ reduce to the case of modifying $o_i$. So this completes the proof. $\square$

**Theorem G.2.** *The sample array $S$ output by Algorithm 6 has the following guarantees:*

1. *Queries are selected uniformly within each group. Formally, consider $i \in I_0$, the subset of all indices such that the query-answer tuple $(q_i, r_i, o_i) \in Q$ has sensitive attribute $s_i = 0$. Then*

$$\Pr[S[i] = 1] = \Pr[S[j] = 1] \quad \forall i, j \in I_0.$$

   *And similarly for queries with sensitive attribute 1.*

2. *Exactly $\nu$ queries in group $a$ and $\nu$ queries in group 1 are included in the sample, for a total of $2\nu$ selected queries. Formally, entries in $S$ are binary-valued, with $\sum_{i \in I_0} S[i] = \nu$. And similarly for queries with sensitive attribute 1.*

3. *(Informal) Algorithm 6 reveals no information to $\mathcal{V}$ other than $n_0$ and $n_1$, the number of queries made by members of each group.*

*Proof Sketch.* In detail, Algorithm 6 randomly permutes all members of group 0 and places the first $\nu$ of them in the sample, and also randomly permutes the all members of group 1 and places the first $\nu$ of them in the sample. To accomplish this operation in zero-knowledge on a committed array of queries, $\mathcal{V}$ constructs two random permutations $\pi_0, \pi_1$ of the appropriate sizes and sends them to $\mathcal{P}$. Each permutation is then loaded into a read-only ZKRAM (see [17] for details), which will allow $\mathcal{P}$ to read permuted elements from $\pi_0$ in the case that they have sensitive attribute $s = 0$ and $\bot$ otherwise (and similarly for group 1), without revealing to $\mathcal{V}$ which case is occurring. We use $\bot$ as a symbol to represent the fact that a query is outside of the group relevant to a permutation.

In lines 4-7, the parties perform a linear traversal of all queries in $Q$, updating a committed counter $[\![c_0]\!]$ whenever a query from group 0 is encountered, and either placing $\pi_0(c_0)$ in an array $A_0$ or $\bot$ depending on group membership via a read to the ZKRAM. At the end of step 10, every query in $Q$ has a corresponding entry in $A_0$ which is $[\![\pi_0(j)]\!]$ if $q_i \in$ group 0, (where $q_i$ is the $j^{th}$ query that belongs to group 0), and $[\![\bot]\!]$ otherwise. We also construct $A_1$ symmetrically for group 1. Thus, each

---

**Algorithm 6:** Group-Balanced Uniform Sample

**Input:**

- The number of queries answered in the Service Phase, $n$, the soundness parameter, $\nu$, which controls how many queries should be sampled per group;
- $\mathcal{P}$ holds an $n$-sized array of committed client queries $[\![Q]\!]$, where $n_0$ entries are from clients in Group 0, and $n_1$ are from clients in Group 0, with $n_0 + n_1 = n$; and
- $\mathcal{V}$ holds commitments to the queries $[\![Q]\!]$.

**Output:** $\mathcal{V}$ and $\mathcal{P}$ respectively receive commitments and committed values for an array of indicator bits $[\![S]\!]$ that encode which values of $Q$ are selected in the sample.

1 $\mathcal{P}$ sends $n_0, n_1$ to $\mathcal{V}$;
2 $\mathcal{V}$ sends $\mathcal{P}$ $\pi_a$, a random permutation of the integers in $[1, n_0]$, and $\pi_b$ a random permutation of $[1, n_1]$;
3 $\mathcal{P}$ loads $\pi_a$ into a read-only ZKRAM $R_a$ such that $R_a[i] = [\![\pi_a(i)]\!]$ $\forall i \in [1, n_0]$. Set $R_a[0] = [\![\bot]\!]$. $\mathcal{P}$ loads $\pi_b$ into $R_b$ similarly, and sets $R_a[0] = [\![\bot]\!]$.;
4 $\mathcal{P}$ commits to a group counter $[\![c_0]\!]$ and initializes it to 1. $\mathcal{P}$ initializes a group-specific permutation array $A_a$ with $n$ entries. **for** $i \in [1, n]$ **do**
5 $\quad$ $[\![b_a]\!] \leftarrow$ indicator bit for $[\![q_i.\texttt{demographic\_attribute}]\!] == a$;
6 $\quad$ $[\![i']\!] \leftarrow [\![b_a]\!] \cdot [\![c_0]\!]$ ; $\qquad\qquad\qquad\qquad \triangleright$ $c_0$ if $q_i$ is in group $a$, 0 otherwise.
7 $\quad$ $A_a[i] \leftarrow R_a[[\![i']\!]]$; $\qquad\qquad\qquad\qquad \triangleright$ $[\![\pi_a(c_0)]\!]$ if $q_i$ is in group $a$, $[\![\bot]\!]$ otherwise.
8 $\quad$ $[\![c_0]\!] \leftarrow [\![c_0]\!] + [\![b_a]\!]$
9 Obtain a group-specific permutation array $A_b$ by repeating lines 4-7 but replacing Group $a$ with Group $b$.;
10 Initialize an array $S$ of size $n$ containing bits indicating whether query $q_i$ is in the sample.;
11 **for** $i \in [0, n)$ **do**
12 $\quad$ $S[i] \leftarrow [\![A_a[i] < \nu]\!] \mid [\![A_b[i] < \nu]\!]$ ; $\triangleright$ $q_i$ is in sample if it's a member of Group $a$ and $\pi_a(i) < \nu$, or it's a member of Group $b$ and $\pi_b(i) < \nu$.
13 return $S$.

---

member of group 0 is labeled with an output from the random permutation $\pi_0$, and each member of group 1 is labeled with an output from $\pi_1$. Then to determine whether a query $q_i$ is included in the uniform sample, lines 10-11 perform one more linear pass over $Q$ wherein a committed sample bit is set to 1 if it's a member of group 0 and $\pi_0(i) < \nu$, or if it's a member of group 1 and $\pi_1(i) < \nu$ (we define $(\bot < k) = 0$ for any $k$).

The construction thus guarantees claims (1) and (2) since group 0 queries with $S[i] = 1$ are exactly those which are mapped to $[0, \nu)$ by the random permutation $\pi_0$, and group 1 queries with $S[i] = 1$ are exactly those which are mapped to $[0, \nu)$ by $\pi_1$, which is equivalent to taking a $\nu$-sized uniform sample from each group. Claim (3) is guaranteed by the underlying ZKP protocols used to perform the authenticated operations and ZKRAM access [47, 17]. $\qquad\square$

**Corollary G.1.** *(Informal) Algorithm 2 reveals no information to $\mathcal{V}$ about the client data and model parameters other than what can be inferred from $n_0$ and $n_1$, and the sample array $S$.*

*Proof Sketch.* By Theorem G.2, taking the group-balanced sample with Algorithm 6 reveals $n_0, n_1$ to $\mathcal{V}$, but no other information. By the security of the underlying ZKP protocol [47] and hash-based commitment scheme [48], all lines besides line 14 leak no additional information to $\mathcal{V}$.

In line 14, after obtaining the sample array $S$, $\mathcal{P}$ opens each of the $2\nu$ bits of $S$ such that $[\![S[i]]\!] == 1$ which effectively reveals $S$ since all other entries are guaranteed to be 0. $\qquad\square$

Revealing $S$ to $\mathcal{V}$ gives some knowledge of which of their commitments correspond to clients included in the sample. By itself, this is an innocuous piece of information – it simply gives $\mathcal{V}$ knowledge that a small subset of query commitments correspond to users with $\frac{1}{2}$ probability of being from group 0 or group 1, rather than $\frac{n_0}{n}$ and $\frac{n_1}{n}$ as is implied by knowledge of $n_0$, $n_1$, and $n$. If we consider stronger adversarial capacities of $\mathcal{V}$, e.g. that they somehow have outside knowledge about $2\nu - 1$ of the queries in the sample, more precise information can be inferred. In this example, $\mathcal{V}$ can infer the

**Algorithm 7:** Fairness Audit w/o $S$ Reveal.

**Input:**
- Public values: $n$ is the number of client queries, $\theta$ thresholds the acceptable level of group fairness gap, and $\nu$ is a soundness parameter.
- $\mathcal{P}$ holds a committed model $M$, and a set of queries authenticating values $(q_i, r_i, o_i) \in Q$ with $i \in [1, n]$.
- $\mathcal{V}$ holds hash-based commitments $C_i = H(q_i||r_i||o_i)$ with $i \in [1, n]$.

// Step 1: Initialization
1 **for** $i \in [1, n]$ **do**
2 $\quad\lfloor\quad$ $\mathcal{P}$ authenticates $(\llbracket q_i \rrbracket, \llbracket r_i \rrbracket, \llbracket o_i \rrbracket)$;

3 $\mathcal{P}$ authenticates $\llbracket M \rrbracket$;
4 $\mathcal{P}$ authenticates $\llbracket c_a \rrbracket$ and $\llbracket c_b \rrbracket$ initialized to zero; $\qquad\triangleright$ Count positive outcomes in each demographic group
5 $\mathcal{P}$ authenticates $\llbracket n_a \rrbracket$ and $\llbracket n_b \rrbracket$ initialized to zero; $\qquad\triangleright$ Count individuals in each demographic group

// Step 2: Measuring Group Fairness
6 **for** $i \in [1, n]$ **do**
7 $\quad\mid\quad$ $\llbracket s_i \rrbracket \leftarrow \llbracket q_i.\text{demographic\_attribute} \rrbracket$;
8 $\quad\mid\quad$ $\llbracket b_a \rrbracket \leftarrow (\llbracket s_i \rrbracket == a), \quad \llbracket b_b \rrbracket \leftarrow (\llbracket s_i \rrbracket == b)$; $\qquad\triangleright$ Indicator bit for demographic attribute
9 $\quad\mid\quad$ $\llbracket n_0 \rrbracket \leftarrow \llbracket n_0 \rrbracket + \llbracket b_a \rrbracket, \quad \llbracket n_1 \rrbracket \leftarrow \llbracket n_1 \rrbracket + \llbracket b_b \rrbracket$; $\qquad\triangleright$ Update group counts
10 $\quad\mid\quad$ $\llbracket c_0 \rrbracket \leftarrow \llbracket c_0 \rrbracket + (\llbracket b_a \rrbracket \cdot \llbracket o_i \rrbracket), \quad \llbracket c_1 \rrbracket \leftarrow \llbracket c_1 \rrbracket + (\llbracket b_b \rrbracket \cdot \llbracket o_i \rrbracket)$; $\quad\triangleright$ Update positive outcome counts

11 $\mathcal{P}$ shows that demographic parity gap is underneath threshold $\theta$ by proving

$$\llbracket b_{\text{pass}} \rrbracket \leftarrow \left( \theta \geq \left| \frac{\llbracket c_0 \rrbracket}{\llbracket n_0 \rrbracket} - \frac{\llbracket c_1 \rrbracket}{\llbracket n_1 \rrbracket} \right| \right)$$

// Step 3: Validity Checks
12 $S \leftarrow$ Group-Balanced Uniform Sampling$(Q, \nu)$; $\qquad\triangleright$ An array indicating selected samples (Algorithm 6)
13 Initialize committed counter $\llbracket x \rrbracket$ to 1;
14 $\mathcal{V}$ publishes all $C_i$, $\mathcal{P}$ loads them into a read-only ZKRAM [17] $Z$;
15 Initialize a read/write ZKRAM [17] $R$ with $(2\nu + 1) \cdot sz$ entries, where $sz$ is the number of values required to store a query-answer tuple $(q, r, o)$ plus its index $i \in [1, n]$. Initialize the tuple-sized block starting at $R[0]$ with $\perp$;
16 **for** $i \in [1, n]$ **do**
17 $\quad\mid\quad$ $\llbracket b \rrbracket \leftarrow S[i]$;
18 $\quad\mid\quad$ $R[\llbracket x \rrbracket \cdot \llbracket b \rrbracket] \leftarrow$ first value in $(q_i||r_i||o_i)||i$, $R[\llbracket x \rrbracket \cdot \llbracket b \rrbracket + 1] \leftarrow$ second value in $(q_i||r_i||o_i)||i$, $\quad\quad$ ..., $R[\llbracket x \rrbracket \cdot \llbracket b \rrbracket + sz] \leftarrow sz^{th}$ value in $(q_i||r_i||o_i)||i$;
19 $\quad\mid\quad$ $\llbracket x \rrbracket \leftarrow \llbracket x \rrbracket + \llbracket b \rrbracket$.

20 **for** $i \in [1, 2\nu]$ **do**
21 $\quad\mid\quad$ $(\llbracket q \rrbracket, \llbracket r \rrbracket, \llbracket o \rrbracket, \llbracket \text{ind} \rrbracket) \leftarrow$ load from $R[i]$;
22 $\quad\mid\quad$ $\mathcal{P}$ proves $\llbracket o \rrbracket == M(\llbracket q \rrbracket, \llbracket r \rrbracket)$ using $\mathcal{F}_{\text{inf}}$;
23 $\quad\mid\quad$ $\mathcal{P}$ proves that $H(\llbracket q \rrbracket || \llbracket r \rrbracket || \llbracket o \rrbracket) == Z(\llbracket \text{ind} \rrbracket)$.

24 If any of the proofs fail, abort. Otherwise, Reveal$(\llbracket b_{\text{pass}} \rrbracket)$.

---

demographic attribute of the single unknown query since they know which group has $\nu - 1$ sampled queries and which group has $\nu$.

While it is important to be mindful of this possible inference, in the proposed setting of a regulatory body auditing the fairness of a machine learning model we anticipate a threat model much closer to the former case than the latter. Further, by integrating a read/write ZKRAM [17] we can remove this leakage in exchange for a minor to moderate additional computational cost (depending on the size of the query input). This is demonstrated in Algorithm 7.

# H  Datasets

Table 4 describes five standard fairness benchmarking datasets used in this paper: COMPAS [2], Communities and Crime [36], Default Credit [51], Adult Income [3], and German Credit [23].

Table 4: Summary of datasets.

| Dataset | License | #Samples | #Attr. | Demographic Var. | Task |
|---------|---------|----------|--------|------------------|------|
| COMPAS | DbCL 1.0 | 6,151 | 8 | Race | Recidivism |
| Crime | CC BY 4.0 | 1,993 | 22 | Race | Crime rate |
| Credit | CC BY 4.0 | 30,000 | 23 | Age | Card Payment |
| Adult | CC BY 4.0 | 45,222 | 14 | Gender | Income |
| German | CC BY 4.0 | 800 | 61 | Foreign Worker | Loan |

# I  Supplementary Results

## I.1  Details of Audit Phase Ablation Study

**Runtime Profiling for Total Query Volume**. OATH computes fairness on all user queries, and uniformly samples $\nu$ queries from each subpopulation for correctness and consistency checks. Both sampling runtime (Algorithm 6) and fairness computation runtime (Algorithm 2 lines 6-11) are independent of the number of sampled queries and the model type – they depend only on the number of user queries $|Q| = n$. Therefore, we show their relationship to the number of queries in Figure 2 (left column). In general, the runtime of both Group-Balanced Uniform Sampling and Fairness Computation increases linearly with the number of user queries. However, the slope of the increase varies across different parts. Given a specific number of user queries, the runtime of fairness computation is significantly lower than the runtime of group-balanced uniform sampling. This is because Algorithm 6 utilizes a ZKRAM primitive [17] in order to realize the group-balanced uniform sample securely, while the fairness computation in Algorithm 2 uses lighter-weight ZK arithmetic operations [47].

**Runtime Profiling for Verified Queries**. OATH verifies user queries through correctness (line 20 in Algorithm 2) and consistency (line 19 in Algorithm 2) checks. Figure 2 (right column) shows the runtime of these operations. Both correctness and consistency checks have time complexity linear in the number of verified client queries. The runtime of the consistency check dominates the overall runtime because it relies on a scheme for converting hash-based commitments to IT-MAC-based [33, 14] authenticated values, which is a relatively expensive operation [48].

## I.2  Additional Details for Parameterizing the Number of Verified Queries

Figure 5 (left column) shows the probability of catching cheating with respect to the number of verified user queries ranging from 1,000 to 10,000. The more queries are verified, the higher chance that cheating is caught. We consider various values for $\epsilon$ – this is the change that $\mathcal{P}$ seeks to induce between the fairness gap measured by the audit, and the true fairness gap during the Service Phase (Definition 4.1). The more unfair $\mathcal{P}$'s model is, the higher they need to set $\epsilon$ to pass the audit. Meanwhile, the probability of evading detection for cheating decreases exponentially as $\epsilon$ gets bigger (Figure 5 right column).

## I.3  Additional Benchmarks

Figure 6 shows the accumulative runtime of computing fairness and checking the correctness of the model using all five datasets.

# J  Limitations

In this study we present OATH, the first method for privacy-preserving certificates of online group fairness. In particular, we use zero-knowledge proofs which make it possible to measure fairness directly from evaluation data, thus removing onerous resource requirements for third-party auditors and the generalization error incurred by using validation sets. Further, our method makes dramatic improvements to scalability in comparison to baselines.

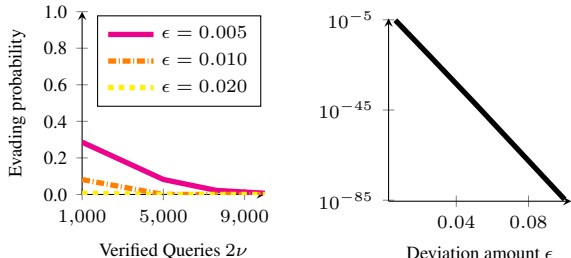

Figure 5: The probability that a cheating $\mathcal{P}$ evades detection when: (Left) deviating from the fairness measurement by three fixed $\epsilon$ values at varying numbers of verified queries; (Right) deviating with 7600 verified queries with varying $\epsilon$. As $\epsilon$ gets smaller, the probability of evasion gets higher but it becomes substantially less impactful on the audit.

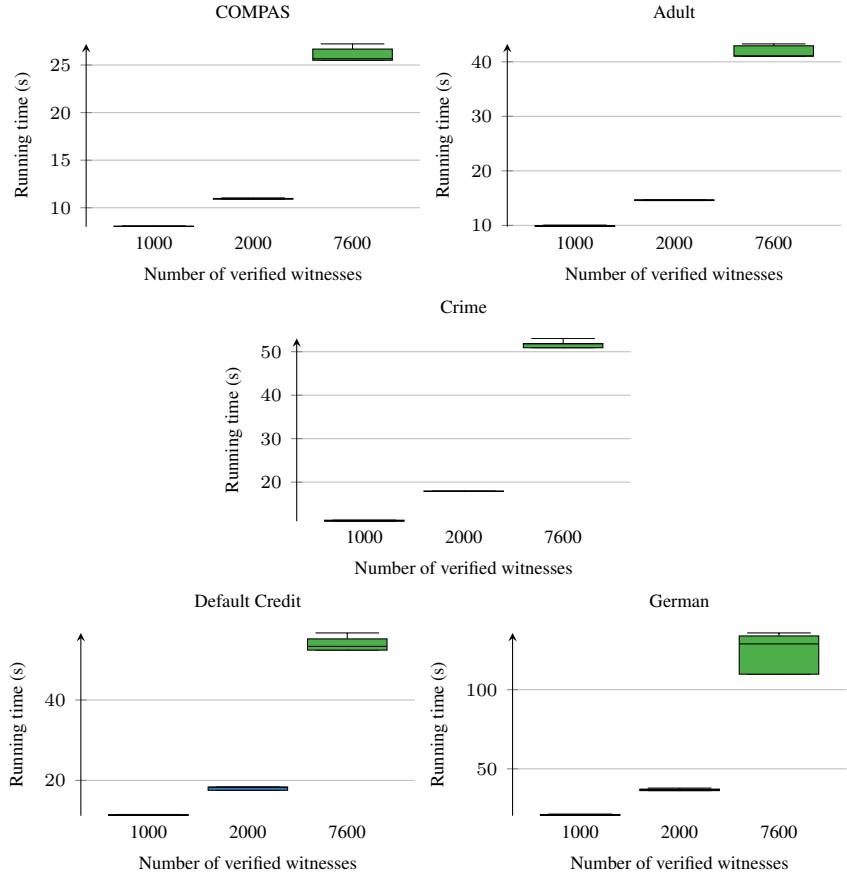

Figure 6: Effect of the number of verified witnesses on the runtime of computing fairness and verifying the inference using the correct model.

While OATH improves the scalability of ZKP fairness auditing by orders of magnitude, the computational requirements could still be an obstacle to adoption in applications that use very large models. Improving the consistency check which comprises our runtime bottleneck is thus an interesting future direction. Of note, our work uses ZKPs of inference (e.g. [48, 10, 42]), and thus will automatically benefit from ongoing advancements in this area.

We also note that while verification of group fairness is important, the metric itself has limitations. Machine learning models can cause inequitable treatment of minority populations even while satisfying group fairness, and thus it is important to not consider cryptographic verification of group fairness as a satisfactory end goal in isolation. Rather, we intend that OATH is used as part of an ensemble of auditing approaches in order to achieve more equitable outcomes from ML tools. To realize this, it may be of interest to expand our cut and choose methodology as a more general framework for efficient ZKP-based ML auditing. Extending this design pattern to wider classes of ML models, fairness metrics, and other trustworthy properties (such as reliability, interpretability, and privacy) is a promising avenue for future work.

Lastly, we comment that while OATH provides security against malicious service providers, auditors, and clients individually, it does not account for collusion between parties. 'Sybil' attacks, wherein a malicious service provider produces fake clients, could be used in a real world attack to make a model appear fair on evaluation data. This could be handled by authentication of clients by the auditor, but it is worth noting as a potential real world attack.

## K  Impact Statement

The present study proposes a method for auditing ML systems for fairness. It is the hope of the authors that OATH and other ZKP-based auditing techniques can be used in service of making deployed ML systems function more ethically by providing accountability for equitable treatment of disadvantaged populations. We stress that OATH and other ZKP-based methods should not be seen as a *replacement* for other forms of auditing. Mathematical fairness criteria alone are not sufficient to determine whether an ML system is functioning ethically - a human needs to be in the loop somewhere. Rather we see these methods as a pathway for providing accountability and transparency in ML systems, to be used as a complement with other methods, so that AI can be regulated more effectively and with lesser expenditure of resources.

