# OpenReview forum: "Secure and Confidential Certificates of Online Fairness"
_NeurIPS.cc/2025/Conference — NeurIPS 2025 poster_

### Official Review · Reviewer_UnaJ · 2025-07-02

**Clarity:** 3
**Significance:** 3
**Originality:** 2
**Rating:** 4
**Confidence:** 2

**Summary:**

This paper examines the problem of certifying online group fairness while maintaining data privacy. Ensuring fairness in machine learning models is particularly important when these models process sensitive user data, yet auditing fairness for confidential datasets remains a significant challenge. To address this, the authors propose the OATH protocol, an efficient and scalable approach based on zero-knowledge proofs for real-time fairness certification. Empirical results demonstrate the protocol's effectiveness in practical settings.

**Questions:**

See "Strengths And Weaknesses".

**Ethical Concerns:**

["NO or VERY MINOR ethics concerns only"]

**Final Justification:**

I standby my initial review.

**Limitations:**

No.

**Paper Formatting Concerns:**

No.

**Quality:**

3

**Strengths And Weaknesses:**

1. Clarity: Clear.
2. Significance: The group fairness and privacy are very important for serving machine learning models.
3. Originality: I am not very familiar with the literature. Looks original to me.
4. Quality: The overall quality seems good. A few things that I hope authors can clarify:
    1. The overall algorithm seems to apply classic zero-knowledge proof algorithm to the setting between a prover and verifier which would like to certify the group fairness gap is bounded by a small value. Could you further clarify why this is not a trivial combination?
    2. This paper did not discuss how floating number is handled. Do you truncate them to fixed point? The fairness gap in Definition 4.1 involves the difference between quantities computed from two groups. We only know the magnitude of quantities after the model is deployed, which may be challenging for fixed point?

---

> ### Author Rebuttal · Authors · 2025-07-31
>
> We thank the reviewer for noting the strength of our paper. We respond below to both questions raised.
>
>
> > **The overall algorithm seems to apply classic zero-knowledge proof algorithm to the setting between a prover and verifier which would like to certify the group fairness gap is bounded by a small value. Could you further clarify why this is not a trivial combination?**
>
> We emphasised that our approach is indeed non-trivial, and we explicitly considered the baseline you suggested – a classic combination of ZKP with fairness verification.
>
> **Why the problem is non-trivial:**
>
> Securely auditing group fairness requires simultaneously satisfying two critical security properties:
>
> 1. query authentication from clients, and
> 2. output authentication from the model owner.
>
> (1) is necessary to prevent model owners from cherry-picking data, which makes their model appear fair, and (2) is necessary to prevent malicious clients from interfering with the auditing process by presenting false outputs.
>
> Ensuring both while keeping the protocol efficient and scalable, especially for large neural networks, poses a significant technical challenge.
>
> **Why the classic ZKP-based approach is insufficient:**
>
> As you suggested, one might attempt to apply a classic ZKP framework to prove correct inference on every query and then prove that the fairness metric holds over all queries. We implemented this baseline (Lines 260-262). As shown in Figure 4, applying classic ZKP to the setting introduces a huge increase in client-facing runtime, computationally infeasible for practical deployment. This is because this naive combination of fairness computation and ZKPs requires proof of correct inference for every user query.
>
> **Our approach – why it's non-trivial:**
>
> Our approach, OATH, introduces a novel cut-and-choose technique that exploits statistical properties of group fairness. OATH achieves strong bounds on fairness violation while only verifying a constant number of user queries (with respect to the total number of queries).
>
>
>
> > **This paper did not discuss how floating number is handled. Do you truncate them to fixed point? The fairness gap in Definition 4.1 involves the difference between quantities computed from two groups. We only know the magnitude of quantities after the model is deployed, which may be challenging for fixed point?**
>
> We use the floating point number implementation in EMP-toolkit. EMP-toolkit supports floating point numbers by representing each float as a list of 32 authenticated bits, and then computing floating point operations as boolean circuits evaluated via the zero-knowledge proof protocol in [47].

---

> > ### Comment · Area_Chair_Vwda · 2025-08-05
> >
> > Hi Reviewer UnaJ,
> >
> > Thank you for your contributions to NeurIPS so far! What do you think about the authors' rebuttal? Did it resolve any of your concerns?
> >
> > Many thanks,
> >
> > Area Chair

---

> > > ### Author Response · Authors · 2025-08-05
> > >
> > > The recent action item sent to all authors (“Discussion period reminder and extension”) recommends that authors initiate discussions with reviewers who have not yet responded to the rebuttal. So we are writing to echo the AC’s comment and kindly request further discussion.
> > >
> > >
> > > We thank the reviewer for recognizing the importance of our topic. As stated in our rebuttal, we believe that our method is a critical improvement upon the trivial combination of zero-knowledge proofs and group fairness verification, which would be far too inefficient to work in practice, especially on larger models. We look forward to further discussion, and we are happy to provide further explanation and/or update our proposed revisions if there are outstanding issues.
> > >
> > >
> > > Sincerely,
> > >
> > >
> > > Authors of Submission 23849

---

> > ### Comment · Reviewer_UnaJ · 2025-08-06
> >
> > The response looks good to me. But I am not an expert in this area and hope Area Chair can take this into account.

---

### Official Review · Reviewer_oaNR · 2025-07-03

**Clarity:** 3
**Significance:** 3
**Originality:** 2
**Rating:** 4
**Confidence:** 4

**Summary:**

The authors discuss the fairness verification of black-box model with confidentiality preservation, which is a significant research topic, since security remains a critical issue in artificial intelligence. The authors proposed to employ zero-knowledge proof method to achieve this goal as well scalability and efficiency. Several experiments are performed to demonstrate its effectiveness and scalability.

**Questions:**

1. Does the experiment include the evaluation of some real models, like CNN or GNN? How about the parameter number influence the method performance?
2. Secure auditing is a classical problem in information security, and fairness auditing of machine learning models can be considered as a special secure auditing problem. Are there any specific challenges faced by fairness verification when compared with the traditional secure auditing problem?
3. It seems that there are some key references not discussed in the work:
[1] FairZK: A Scalable System to Prove Machine Learning Fairness in Zero-Knowledge
[2] Oath: Efficient and flexible zero-knowledge proofs of end-to-end ml fairness
Please have a detailed comparison to illustrate the difference between the proposed work and these references.

**Ethical Concerns:**

["NO or VERY MINOR ethics concerns only"]

**Final Justification:**

My original concern mainly lies in two points. The first one is that the fairness verification issue discussed in this paper can be considered as a secure auditing problem, which has been widely discussed in the field of information security. The proposed method is not directly related to the model structure, but works on the model output. Nevertheless, after reading the response from the authors and the discussion of other reviewers, I consider that the efficient fairness verification method in this work is of significance and indicates contributions. Thus, my first concern is addressed. The second concern is that the mentioned arXiv paper indicates certain similarities to this one, including research problems, methods, performance, and figures, and that is why I expect the authors to discuss the work as well and provide some comparison. However, this concern is not currently addressed.

Therefore, considering that my concerns are partially addressed, I raise the rating from 3 to 4.

**Limitations:**

Refer to the weakness.

**Paper Formatting Concerns:**

No major formatting issues.

**Quality:**

2

**Strengths And Weaknesses:**

Strengths:

1. The work is well-written, and the research topic is significant.
2. The work is technically solid, and zero-knowledge proof can well address the target issue;
3. Several analyses and experiments are performed to demonstrate the scalability and security of the work.

Weaknesses:
1. The fairness should be conceptually explained before the formal definition.
2. It seems the work does not fully review existing works, and more analyses should be added.
3. Secure auditing is a classical problem in information security, and fairness auditing of machine learning models can be considered as a special secure auditing problem. Are there any specific challenges faced by fairness verification when compared with the traditional secure auditing problem?
4. It seems that there are some key references not discussed in the work:
[1] FairZK: A Scalable System to Prove Machine Learning Fairness in Zero-Knowledge
[2] Oath: Efficient and flexible zero-knowledge proofs of end-to-end ml fairness
Please have a detailed comparison to illustrate the difference between the proposed work and these references.

---

> ### Author Rebuttal · Authors · 2025-07-31
>
> Thank you for the review.
>
> > **Fairness should be conceptually explained before the formal definition.**
>
> We will improve our exposition on fairness by revising the Fairness subsection in the preliminaries, and adding the following text:
>
> “Demographic parity requires that the probability of a positive outcome for each demographic subgroup is nearly equal. For instance, if an ML model assists a bank with allocating loans, the model should give a male clients loans about as often as it gives them to female clients.”
>
> > **Does the experiment include the evaluation of some real models, like CNN or GNN? How about the parameter number influence the method performance?**
>
> Yes, our experiments in the paper include evaluations of real CNN models such as ResNet-101 (42.5 million parameters) and ResNet-50 (23.5 million parameters). Our design enables “plug and play” fairness auditing for any classifier with a zero-knowledge proof of correct inference.
>
> Figure 4.b illustrates the impact of model parameters on the method performance. The results indicate that the online runtime (i.e., service phase runtime) is independent of the number of model parameters. However, the audit runtime increases as the number of model parameters increases, as the correctness check (Algorithm 2 line 20) requires a ZKP correct inference of the model, which scales with the model size.
>
>
> > **Secure auditing is a classical problem in information security, and fairness auditing of machine learning models can be considered as a special secure auditing problem. Are there any specific challenges faced by fairness verification when compared with the traditional secure auditing problem?**
>
> Most works in the literature that use Secure Auditing as a keyword (e.g. [a] “Secure auditing and deduplicating data in cloud” and [b] “Dynamic-hash-table based public auditing for secure cloud storage”) are concerned with verifying information integrity in cloud storage. Using these methods directly would be insufficient to verify the integrity of the predictions from the service provider’s ML model, though adapting them to efficiently verify the database of client queries $Q$ could be an interesting direction for future work.
> The key difference in the settings is that if we do not verify the integrity of the model prediction output by the service provider, i.e. that $M(q_i)=o_i$, then the service provider is free to switch models or indeed output arbitrary decisions, rendering the results of the fairness audit unreliable for future clients. This is why we require zero-knowledge proofs of ML inference as a component of OATH.
>
> Would the reviewer please verify whether this is their intended meaning when referring to secure auditing as a classical problem in information security?
>
> > **Related work.**
>
> We compared against existing published fairness auditing approaches, namely Black-Box audits [34, 35, 37] as well as both previous works that leveraged ZKPs to enforce model fairness (FairProof [49]  and Confidential-PROFIT [40]). We also created a baseline based on ZKP of correct inference, Mystique [48].
>
> The ref [1] suggested by you is a preprint that was submitted to arxiv on May 12, 2025 – only three days before the NeurIPS submission deadline (May 15, 2025). Given this it was not feasible to discuss it in our manuscript. We will add the following to our revision:
>
> "In concurrent work, [1] proposed a new zkSNARK-based proof of fairness. Their approach is highly efficient, but relies on proving bespoke bounds for each new model type, unlike OATH which can "plug and play" to automatically prove fairness for any model with a zero-knowledge proof of inference. Thus we are able to evaluate our method on more complex architectures such as convolutional neural networks, while their study is limited to logistic regression and multi-layered perceptrons."

---

> > ### Comment · Reviewer_oaNR · 2025-08-05
> >
> > Thanks for your careful response, which has addressed some of my previous concerns. However, some questions still remain unanswered. I have suggested two references, but it seems that the authors only discussed the first one. The second one indicates some similarity with this work, which is an arXiv paper submitted in 2024:
> >
> > **Olive Franzese, Ali Shahin Shamsabadi, Hamed Haddadi, "OATH: Efficient and Flexible Zero-Knowledge Proofs of End-to-End ML Fairness," https://arxiv.org/abs/2410.02777v1.**
> >
> > Please have a discussion with the proposed work with this reference. I will consider adjusting my rating after it.

---

### Official Review · Reviewer_64AH · 2025-07-05

**Clarity:** 3
**Significance:** 3
**Originality:** 3
**Rating:** 5
**Confidence:** 3

**Summary:**

The presented work introduces ​online fairness certificates​ using zero-knowledge proofs (ZKPs), addressing critical limitations of offline fairness certification. The problem arise in scenarios where both fairness in life-changing tasks (loan approval and recruitment) and data privacy & model IP are to be protected.

This paper proposed to measure fairness directly from the client data received online during deployment, and preserve confidentiality by issuing online fairness certificates with zero-knowledge proofs (ZKPs).

**Questions:**

The paper is well-organized and presented. Nevertheless, I do have few questions/suggestion concerning the experiments and applicability of the method.

1. Any other baseline methods to be compared, e.g. in terms of the trade-off of relaibility / efficiency ?
2. it would be good to empirically demonstrate the defense capability of the proposed method e.g. under various attacks (data-forging or model-swtiching).

And at last, have you ever considered to apply the same approach to address other issues e.g. online-security certificate?

**Ethical Concerns:**

["NO or VERY MINOR ethics concerns only"]

**Limitations:**

yes

**Paper Formatting Concerns:**

no concerns

**Quality:**

3

**Strengths And Weaknesses:**

+ the notion of online group fairline certificate is well-defined and presented.
+ a reliably and scalable solution is provided s.t. third-party verifier can be convinced.

- it seems only one baseline method is compared in empricial evaluation, while FairProof is only compared for computatoinal overhead. It would be good to present more emprical results.

---

> ### Author Rebuttal · Authors · 2025-07-31
>
> We thank the reviewer for the positive assessment of our paper.
>
> > **Any other baseline methods to be compared, e.g. in terms of the trade-off of relaibility / efficiency?**
>
> We compared against existing fairness auditing approaches, namely Black-Box audits [34, 35, 37] as well as both previous works that leveraged ZKPs to enforce model fairness (FairProof [49]  and Confidential-PROFIT [40]). We also created a baseline based on ZKP of correct inference, Mystique [48].
> We will clarify these categories in the introduction as:
>
> “Black-box fairness auditing considers an auditor who has only query access to a model, and so measures fairness based on submitted queries and observed decisions of the service provider. By contrast, white-box fairness auditing approaches release client data and the service provider’s model to the auditor, who can then verify fairness. Both approaches have drawbacks: the white-box setting is seldom used in practice due to issues with confidentiality, while the black-box setting does not provide the auditor enough information for a rigorous audit [9] (for example, nothing prevents a service provider from using one model during the audit, and a completely different one to answer client queries).
>
> Zero-knowledge proofs of fairness [40, 49] are an emerging line of work that address both of these weaknesses using cryptographic tools. Zero-knowledge proofs essentially provide secure “white-box” access to the auditor for a set of pre-specified operations, while retaining (and in fact improving upon) the confidentiality of the black-box setting.”
>
> We will add a comparison of all these fairness auditing approaches in terms of the trade-off of reliability / efficiency:
>
> | **Approach**        | **Fairness** | **Certificate** | **Reliable**                         | **Efficient** |
> |---------------------|--------------|------------------|--------------------------------------|---------------|
> | Black-Box [34,35,37]  | Various      | Online        | -                      | ✓             |
> | Confidential-PROFIT [40]      | Group        | Offline         | -                    | -             |
> | FairProof [49]      | Individual   | Online        | ✓          | -             |
> | Fairness Baseline w/ Mystique [48]      | Various   | Online        | ✓          |  -            |
> | OATH            | Group    | Online   | ✓     | ✓         |
>
>
>
> > **it would be good to empirically demonstrate the defense capability of the proposed method e.g. under various attacks (data-forging or model-swtiching).**
>
> Defence capability of the proposed method against
> - Model switching attack: Model switching attacks are fully mitigated by the zero-knowledge proof of correct inference included in our protocol. This is because the proof verifies that the inference was run on the committed model (Line 20 in Algorithm 2).
>
> - Data forging attack: Figure 3 (Section 5.2) and Figure 5 (Appendix H) empirically quantify the probability that a cheating service provider evades the detection component of our protocol when falsely claiming a group fairness threshold $\theta$ by using a forged dataset. These results show that our protocol is robust: a service provider that cheats with a larger than negligible $\epsilon$ will be caught by the auditor with high probability. The reason we have not added more empirical results is that we theoretically establish this robustness in Theorem 4.1 and proved that it only depends on the number of verified queries. However, we are happy to extend our empirical results under additional attack scenarios if the reviewer feels it would strengthen the paper.
>
>
>
> > **have you ever considered to apply the same approach to address other issues e.g. online-security certificate?**
>
> Thank you for the thoughtful suggestion. Fairness has the convenient property of being a measurable function of model input/output pairs, allowing us to utilise evaluation data directly in order to audit fairness.
> That said, we agree that extending this approach to domains like online security is an interesting future direction. We will add the following to our discussion of future work:
>
> “An interesting future direction would be to explore the applicability of ZKP certificates to online-security guarantees.”

---

> > ### Comment · Reviewer_64AH · 2025-08-05
> > **thanks for clarification**
> >
> > well-done, no more questions.

---

### Official Review · Reviewer_7kjt · 2025-07-16

**Clarity:** 2
**Significance:** 2
**Originality:** 2
**Rating:** 4
**Confidence:** 4

**Summary:**

This paper introduces OATH, a method to audit the fairness of machine learning (ML) models in a black-box fashion. The method uses a zero-knowledge proof (ZKP) protocol in order to ensure confidentiality, namely to prevent the auditor from learning the value of the sensitive attribute. The values of the sensitive attribute represent the groups with respect to which fairness is defined.

The paper focuses on the notion of fairness called "statistical parity", which limits the difference between the probability of obtaining a positive prediction across different groups. In the appendix, however, the paper discusses the extension to other kinds of fairness notions.

The protocol operates in two phases:

- Service Phase: the clients query the model and send cryptographic commitments of their queries and results to the auditor.

- Audit Phase: The auditor interacts with the model to verify the validity of the fairness threshold, and then it checks whether there has been tampering on a randomly sampled subset of the queries provided by the clients, chosen to be balanced across the various groups. This subset is of bounded size, which provides scalability of the tampering check. This phase performs various validity checks (sensitive attribute check, commitment consistency check, and inference correctness check)  using ZKPs.

The paper also provides a formal proof that the probability with which a malicious service provider can wrongly demonstrate that its model is fair (by tampering with the data) decreases exponentially as a function of sample size.

**Questions:**

Please clarify whether the sensitive attribute is a characteristic of the client (which is consistent with what is written in Algorithm 1, Line 2), or of the query  (which is consistent with what is written in Algorithm 2, Line 7).

**Ethical Concerns:**

["NO or VERY MINOR ethics concerns only"]

**Final Justification:**

I had initially misunderstood the scenarios that the paper is addressing and the goals. The authors clarified these misunderstandings during the discussion phase, and consequently, I have raised my score.

**Limitations:**

Yes

**Paper Formatting Concerns:**

No concerns about formatting

**Quality:**

3

**Strengths And Weaknesses:**

I had to re-read the paper several times because, at first, it did not make any sense to me. Then, I think I finally understood that this paper considers the sensitive attribute to be a feature of the client, rather than -- as it is usually assumed -- an attribute within the query.

Under the standard assumption that the sensitive attribute is part of the query (and that a client can deal with queries with different values of the sensitive attribute), all the effort made by this paper to ensure confidentiality is "the classic cannon to kill a fly". A much simpler protocol would be the following: for each query, a client queries the model and then sends the corresponding pair (sensitive attribute_i, prediction_i) to the auditor. Once all these pairs are collected,  the auditor performs a simple calculation to check whether the fairness threshold is respected. There is no substantial confidentiality issue since the pair (sensitive attribute_i, prediction_i) does not leak any significant information about the data, other than there is a data point with that sensitive attribute.

The paper must therefore assume that the sensitive attribute is a characteristic of the client, and that we want to prevent the auditor from learning it. Even then, however, I don't see the need for such complicated protocol. One could just use some protocol for collecting the pairs (sensitive attribute_i, prediction_i) anonymously, for instance, a mixnet protocol, and then send the collection to the auditor.

Furthermore, often the sensitive attribute can be inferred from other attributes in the query (proxies). Hence, protecting only the sensitive attribute, as is done in this paper, would not be sufficient.

The paper is also lacking clarity. For instance, it's not clear which entity performs Steps 6-10 in Algorithm 2: if it is the service provider, then I don't see how we can prevent it from cheating on this point, and simply return a number that verifies the fairness threshold. If it is the auditor, then it learns (in Line 7) the sensitive attribute of the client, since it knows from which client the query $q_i$ comes.

Furthermore, readability is poor due to several undefined notations and notions, and to the presence of some inconsistencies: For instance, what is the meaning of the $\parallel$ symbol in Algorithm 1? What is the meaning of the symbol $\oplus$ in Algorithm 2? What are the values of the sensitive attribute? In Line 78 they are $a,b$, but in Algorithm 2, they seem to be $0,1$.

Some of the claims are misleading. For instance, the paper makes it sound like the proposed zero-knowledge proof protocol protects the confidentiality of the model, namely, it prevents the auditor from learning the parameters of the model, but this is simply a consequence of the fact that the auditing is performed in a black-box fashion.

---

> ### Author Rebuttal · Authors · 2025-07-31
>
> The authors thank the reviewer for this perspective. We posit that weaknesses raised in this review are primarily issues with communication across an interdisciplinary line (between cryptography and machine learning), rather than problems with the merit of the work. The reviewer raises many good points which will help us address those communication issues. We hope that this rebuttal will make our contributions more clear, and that the reviewer will continue in dialogue with us to make the paper stronger and more impactful for all readers.
>
> > **the paper makes it sound like the proposed zero-knowledge proof protocol protects the confidentiality of the model, namely, it prevents the auditor from learning the parameters of the model, but this is simply a consequence of the fact that the auditing is performed in a black-box fashion.**
>
> We clarify the difference between black-box audits and zero-knowledge proofs with the following text, which we will add to our introduction:
>
> “Black-box fairness auditing considers an auditor who has only query access to a model, and so measures fairness based on submitted queries and observed decisions of the service provider. By contrast, white-box fairness auditing approaches release client data and the service provider’s model to the auditor, who can then verify fairness. Both approaches have drawbacks: the white-box setting is seldom used in practice due to issues with confidentiality, while the black-box setting does not provide the auditor enough information for a rigorous audit [9] (for example, nothing prevents a service provider from using one model during the audit, and a completely different one to answer client queries).
>
> *Zero-knowledge proofs of fairness* [40, 49] are an emerging line of work that address both of these weaknesses using cryptographic tools. Zero-knowledge proofs essentially provide secure “white-box” access to the auditor for a set of pre-specified operations, while retaining (and in fact improving upon) the confidentiality of the black-box setting.”
>
> Addressing the reviewer’s point directly: while a black-box audit does indeed keep model parameters confidential, only a zero-knowledge proof can keep model parameters confidential while also guaranteeing that the audit was computed correctly. For example, if we do not use a ZKP to verify that $M(q_i)=o_i$ (i.e. that the model prediction output by the service provider is correctly computed), then the service provider is free to switch models at will or even output fully arbitrary decisions, rendering the results of the fairness audit unreliable for future clients.
>
> > **Why doesn’t the client simply query the model and then send the corresponding pair (sensitive attribute_i, prediction_i) to the auditor?**
>
> This black-box approach is insufficient for two main reasons:
> - *It can be abused by the clients* who may report a prediction to the auditor which is different from the one they received, in order to make the service provider’s model appear unfair. We discuss this possibility in Appendix C: “a malicious [client] may seek retribution for an undesirable model prediction.” We will move this discussion to the main text for clarity.
> - *It can be abused by the service provider* because it offers no way to verify whether the service provider’s output was a model prediction or simply an arbitrary decision. If we do not verify that $M(q_i)=o_i$ (where $q_i$ is the query data of the ith client, and $o_i$ is the prediction given to the ith client) using a zero-knowledge proof of inference, then the service provider is free to switch models or indeed output arbitrary decisions, rendering the results of the fairness audit unreliable for future clients. This opens the door to e.g. fairwashing attacks discussed in previous work [1, 39].
>
> The consensus of many related works in the literature (e.g. [10, 28, 31, 32, 38, 40, 42, 48, 49]) is that cryptographic methods are necessary to close vulnerabilities such as these.
>
>
> > **Is the sensitive attribute a feature of the client (Algorithm 1, Line 2) or of the query (Algorithm 2, Line 7)?**
>
> We thank the reviewer for raising this point. Indeed, in typical fairness literature sensitive attributes are considered a feature of the query only, rather than a feature of the client. We did not distinguish between these two cases, since they are equivalent in all of our experimental evaluations: e.g. in the COMPAS dataset the sensitive attribute is race, so a client’s sensitive attribute would not change for different queries. To align with the previous literature, we will update Algorithm 1, Line 2 to read:
> {$s \gets q.$sensitive_attribute}.
>
> Our implementation handles this without modification.
>
>
> > **the sensitive attribute can be inferred from other attributes in the query (proxies). Hence, protecting only the sensitive attribute, as is done in this paper, would not be sufficient.**
> &
> >  **it's not clear which entity performs Steps 6-10 in Algorithm 2: if it is the service provider, then I don't see how we can prevent it from cheating on this point, and simply return a number that verifies the fairness threshold. If it is the auditor, then it learns (in Line 7) the sensitive attribute of the client, since it knows from which client the query $q_i$ comes.**
>
> It is ***not*** the case that our method protects only the sensitive attribute of the client query. Our method prevents the auditor from learning *any information* about the entire client query, using the security guarantees of zero-knowledge proofs. It also prevents the service provider from cheating. These properties are both guaranteed by our ZKP building blocks. This is discussed in Appendix B, but we provide additional detail here:
>
> - Values appearing in double brackets are *IT-MAC authenticated*. The notation $[[x]]$ means that that the *auditor* holds a global key $\Delta$ and a message key $K_x$, while the *service provider* holds the value $x$ and message tag $M_x$ such that $M_x = K_x + x \cdot \Delta$.
> - Whenever operations on authenticated values are written (e.g. in Steps 6-10 of Algorithm 2), it means that we are using the zero-knowledge proof protocols in [46, 47, 17] to perform those operations securely. The auditor and service provider *work together*, computing distinct operations on their respective pieces of the authenticated values. These result in new authenticated values for the outputs.
>      - The methods for performing secure operations on authenticated values are complicated (see [47]), but we will give a simple example: finding sum of authenticated values $[[z]] \gets [[x]] + [[y]]$. To accomplish this, the auditor computes $K_z \gets K_x + K_y$, while the service provider computes $z \gets x + y$ and $M_z \gets M_x + M_y$. This preserves the relation $M_z = K_z + z \cdot \Delta$ (which can be checked at the end of the protocol to prevent the service provider from cheating by changing the value of $z$) even while the auditor never sees the values $x, y, z$.
> - The security proofs in [47, 17] show that these operations are guaranteed to compute the outputs correctly (via the *Soundness* property of the zero-knowledge proof) and without revealing any information to the service provider (via the *Zero-Knowledge* property).
>
> Thus, cheating and breach of confidentiality are indeed prevented in Steps 6-10 of Alg 2. We will add more detailed exposition to the zero-knowledge proof section of the preliminaries, and add an expanded walk-through example to the Appendix, to make the use of authenticated values more clear.
>
> > **Notation.**
>
> In cryptography literature $||$ and $\oplus$ are standard notation for concatenation and XOR respectively. We will define both explicitly in the revision.
>
> > **What are the values of the sensitive attribute? In Line 78 they are a,b, but in Algorithm 2, they seem to be 0,1.**
>
> We will clarify this by unifying sensitive attribute values to be {0,1} everywhere in our revision.

---

> > ### Author Response · Authors · 2025-08-05
> >
> > The recent action item sent to all authors (“Discussion period reminder and extension”) recommends that authors initiate discussions with reviewers who have not yet responded to the rebuttal. So we are writing to kindly request further discussion.
> >
> >
> > We reiterate that we value this reviewer’s perspective, and that we are invested in addressing the issues raised in order to better communicate the merit of our work to all readers. We would greatly appreciate the reviewer’s input about whether our rebuttal addresses their concerns about notational consistency, explanation of authenticated values, and whether it clarifies the difference between black-box approaches and ZKP-based approaches. We are happy to provide further explanation and/or update our proposed revisions if there are outstanding issues.
> >
> >
> > Sincerely,
> >
> >
> > Authors of Submission 23849

---

> > > ### Comment · Reviewer_7kjt · 2025-08-06
> > >
> > > I'd like to thank the authors for the time and effort spent on the rebuttal to my review, but I still don't agree with their notion of  fairness.
> > >
> > > Concerning my remark about the notion of fairness, they wrote, "We thank the reviewer for raising this point. Indeed, in typical fairness literature, sensitive attributes are considered a feature of the query only, rather than a feature of the client. We did not distinguish between these two cases, since they are equivalent in all of our experimental evaluations: e.g. in the COMPAS dataset, the sensitive attribute is race, so a client’s sensitive attribute would not change for different queries."
> > >
> > > From the above answer, I deduce that the authors think that the client who queries the COMPAS system is the convict himself. This is wrong: it's the Department of Justice that submits the query by giving as input the dossier of the convict.  And in general, the problem of fairness in ML is not that the system produces unfair predictions for the entity querying it. It produces unfair predictions for some groups whose identity is somehow revealed by the attributes in the query.
> > >
> > > Also, I don't agree with their reply to my remark that the system can infer the value of the sensitive attribute from other attributes in the query. They wrote: "Our method prevents the auditor from learning any information about the entire client query". I think they did not understand what I meant. For instance, in the case of COMPAS, the entity training (resp. querying) the ML model may remove the attribute "race" from the sample  (resp. query), but the model can infer the race, with a good probability to be correct, from other attributes that remain present in the query, for instance, the area in which the convict lives, his surname, etc. This is a well-known problem in the fairness literature.

---

> > > > ### Author Response · Authors · 2025-08-06
> > > >
> > > > Thank you for your continued engagement with our work.
> > > >
> > > >
> > > > > **I still don't agree with their notion of fairness.**
> > > >
> > > >
> > > > Our method audits standard group fairness metrics taken directly from the fairness literature.
> > > >
> > > >
> > > > > **I deduce that the authors think that the client who queries the COMPAS system is the convict himself. This is wrong: it's the Department of Justice that submits the query by giving as input the dossier of the convict. And in general, the problem of fairness in ML is not that the system produces unfair predictions for the entity querying it. It produces unfair predictions for some groups whose identity is somehow revealed by the attributes in the query.**
> > > >
> > > > It was not our intention to imply that defendants directly query the COMPAS system – we used COMPAS as an example of an immutable sensitive attribute.
> > > >
> > > > We would characterize the setting that the reviewer suggests as follows:
> > > > - The proprietary COMPAS system is the service provider $\mathcal{P}$
> > > > - A set of $N$ court offices are the clients $\{ \mathcal{C} \}^N_{i=1}$
> > > > - An independent review body seeking to measure fairness is the auditor $\mathcal{V}$
> > > >
> > > >
> > > > (Please let us know if this is not what you have in mind)
> > > >
> > > >
> > > > Our method supports this setting directly. Each court office collects dossiers from the defendants, and submits them according to Alg 1: $\mathcal{P}$ receives each query (i.e. the features of the dossier, *including* sensitive attribute), and $\mathcal{V}$ receives a cryptographic ‘receipt’ of each query. Then, $\mathcal{P}$ and $\mathcal{V}$ use Alg 2 to prove group fairness across all received queries. This way, (i) the fairness of $\mathcal{P}$’s model is measured correctly -- $\mathcal{P}$ cannot alter their outputs to make the model appear more fair, (ii) intellectual property is protected since neither the court offices nor the review body learn anything about the model, (iii) the review body obtains no information about the defendants, protecting their privacy (important in many real-world settings, where privacy legislation may prevent sharing of data across jurisdictions).
> > > >
> > > > In summary, our method supports *both* settings where clients are entities which aggregate multiple queries, *and* settings where clients are the individuals represented by each query. Please let us know whether this addresses your concern, we are happy to discuss further.
> > > >
> > > >
> > > >
> > > >
> > > > > **For instance, in the case of COMPAS, the entity training (resp. querying) the ML model may remove the attribute "race" from the sample (resp. query), but the model can infer the race, with a good probability to be correct, from other attributes that remain present in the query, for instance, the area in which the convict lives, his surname, etc. This is a well-known problem in the fairness literature.**
> > > >
> > > > We are indeed familiar with the fact that the sensitive attribute can often be predicted from a query. We do not believe that this poses any problem for our method.
> > > >
> > > >
> > > > By design, the service provider $\mathcal{P}$ receives the whole query (all features including sensitive attribute, Alg 1 line 4) just as they do normally in the clear. Thus, inferring the sensitive attribute from other features does not provide $\mathcal{P}$ with any information they did not already have.
> > > >
> > > >
> > > > Perhaps the reviewer is under the impression that the purpose of the $\alpha_s$ value sent in Alg 1 line 4 is to hide the sensitive attribute from $\mathcal{P}$. This is not the case – $\mathcal{P}$ is able to see the sensitive attribute directly, as it is part of $q$. Rather, the purpose of $\alpha_s$ is to enable the sensitive attribute check in the Audit Phase (Alg 2, lines 14-17) which prevents $\mathcal{P}$ from *changing* the sensitive attribute of any query without detection by the auditor. We will add comments in Alg 1, and text in Section 4.1 to clarify this.
> > > >
> > > >
> > > > Meanwhile, $\mathcal{V}$ learns no information about the query since they receive only a cryptographic commitment string and two uniform random bits (Alg 1, line 10). So there is no way they could infer the sensitive attribute from this information.
> > > >
> > > >
> > > > If this does not address the reviewer’s concern, would they please clarify what problem the mentioned fact poses for our framework?

---

> ### Comment · Reviewer_7kjt · 2025-08-07
>
> Thanks for the explanation, the example of COMPAS is very clear now. I had misunderstood and scenarios you are addressing and the goals of your paper, because I am used to a different trade-off between fairness and privacy in the context of ethical AI. I will raise my score.
>
> I suggest that you include the example of COMPAS, in the way you have explained to me, in the revised version.

---

### Comment · Area_Chair_Vwda · 2025-08-03
**Discussion period**

Dear Reviewers,

Thank you so much for all your time and effort supporting NeurIPS!

If you haven't yet, please take a moment to read through the author's rebuttal. If the rebuttal addresses your concerns, please acknowledge this and adjust your scores accordingly. If not, please let them know which concerns remain and if you have any follow-up questions. Your thoughtful feedback is crucial for helping the authors improve their work and advancing the field.

I realize this is a busy time and really appreciate your effort.

Best Regards,

Area Chair

---

### Decision · Program_Chairs · 2025-09-17

**Decision:**

Accept (poster)

**Comment:**

The paper proposes a zero-knowledge method for auditing the fairness of a black-box model. The problem statement is clear and well-motivated, and the authors propose a reasonable algorithm for realizing the stated goals. The reviewers have made multiple suggestions related to improving the clarity of the problem setting and guarantees. These include: (a) clarifying the problem statement and notion of fairness under consideration, and including a concrete example to explain the intended use case, (b) including a qualitative description comparing to other baselines, and also addressing the method's resilience to various attacks. Overall, the reviewers' major concerns were largely addressed during the rebuttal phase. I strongly suggest incorporating these clarifications into the next revision of the paper.